# CRISPR/Cas9-mediated excision of ALS/FTD-causing hexanucleotide repeat expansion in *C9ORF72* rescues major disease mechanisms in vivo and in vitro

A GGGGCC$_{24+}$ hexanucleotide repeat expansion (HRE) in the *C9ORF72* gene is the most common genetic cause of amyotrophic lateral sclerosis (ALS) and frontotemporal dementia (FTD), fatal neurodegenerative diseases with no cure or approved treatments that substantially slow disease progression or extend survival. Mechanistic underpinnings of neuronal death include C9ORF72 haploinsufficiency, sequestration of RNA-binding proteins in the nucleus, and production of dipeptide repeat proteins. Here, we used an adeno-associated viral vector system to deliver CRISPR/Cas9 gene-editing machineries to effectuate the removal of the HRE from the *C9ORF72* genomic locus. We demonstrate successful excision of the HRE in primary cortical neurons and brains of three mouse models containing the expansion (500–600 repeats) as well as in patient-derived iPSC motor neurons and brain organoids (450 repeats). This resulted in a reduction of RNA foci, poly-dipeptides and haploinsufficiency, major hallmarks of C9-ALS/FTD, making this a promising therapeutic approach to these diseases.

Amyotrophic lateral sclerosis (ALS) is a neurodegenerative disease that affects upper and lower motor neurons in the brain and spinal cord. It is characterized, primarily, by progressive muscle weakness that ultimately results in respiratory failure. ALS is a debilitating disease that is invariably fatal with most affected individuals succumbing within 3–5 years of symptom onset[1]. The 2011 discovery of a hexanucleotide repeat expansion (HRE) in the noncoding region of the Chromosome 9 Open Reading Frame 72 (*C9ORF72*) gene revealed this to be the most common known cause of both hereditary (40%) and apparently-sporadic (5–6%) ALS cases[2–4]. In healthy individuals, the size of the hexanucleotide sequence is less than 24 repeats, but affected individuals have expansion mutations that can number in the thousands of repeats[5,6]. The HRE is also the most common genetic cause of frontotemporal dementia (FTD), a related neurodegenerative disease with multiple clinical manifestations, including disinhibited behavior, language disorders, loss of executive control, and motor symptoms[7]. Both ALS and FTD are aggressive diseases with no current treatments that significantly slow disease progression or extend life expectancy[8].

The *C9ORF72* gene locus produces three protein-coding RNA transcript variants, V1, V2, and V3 through alternative splicing and transcription start site use[2]. V1 is translated into a 222 amino acid protein, while V2 and V3 encode the predominant 481 amino acid C9ORF72 protein[2]. C9ORF72 interacts with endosomes and is required for normal vesicle trafficking, autophagy induction and lysosomal biogenesis in diverse cell types, including motor neurons[9,10]. The HRE gives rise to three pathological hallmarks of C9ORF72 ALS: (1) It impairs expression, leading to C9ORF72 haploinsufficiency that compromises neuronal viability[10,11]. (2) Sense and antisense transcription of the *C9ORF72* HRE produces G$_4$C$_2$ and C$_4$G$_2$ transcripts that accumulate in cell nuclei and sequester RNA-binding proteins, resulting in RNA foci[2,12–19]. (3) Sense and antisense HRE transcripts are translated via an abnormal mechanism into toxic poly-dipeptides, producing glycine–arginine (GR), glycine–proline (GP),

e-mail: zzeier@med.miami.edu; Christian.mueller4@sanofi.com

and glycine-alanine (GA) poly-dipeptides in the sense direction and proline–alanine (PA), proline–arginine (PR), and GP in the antisense direction[20–25]. These aggregation-prone poly-dipeptides have been found in the brains and spinal cords of C9ORF72 ALS and FTD patients[17,25,26] and are toxic in cell culture[27–30] and animal models[31–37]. Furthermore, there is a direct form of cooperative pathogenesis between these gain- and loss-of-function mechanisms in C9ORF72, where C9ORF72 haploinsufficiency impairs clearance of poly-dipeptides, making motor neurons hypersensitive to the toxic effects of the poly-dipeptides[10,11]. Despite these advances in elucidating the molecular pathology of the HRE, there are no approved therapies that have been developed based on this knowledge. Existing treatments do not directly target the pathological hallmarks of *C9ORF72* ALS/FTD and are very modestly beneficial[8].

The field of gene therapy and gene editing offers potential treatment modalities for C9-ALS/FTD pathology by attenuating *C9ORF72*-related toxicity at the genetic level. Among these, one attractive approach is to physically excise the repeat expansion mutation from the *C9ORF72* genomic locus using gene-editing tools such as the CRISPR/Cas9 system. The Cas9 endonuclease produces double-strand breaks in genomic DNA at specific sequences that are determined by complementarity with a guide RNA (gRNA) and the presence of a protospacer-adjacent motif (PAM)[38,39]. This versatile genome editing technology is currently being tested in human clinical trials for the treatment of sickle cell disease, transthyretin amyloidosis, and vision loss by Leber's congenital amaurosis[40–42].

Here, we developed an AAV9-mediated CRISPR/Cas9 gene-editing approach to remove the GGGGCC$_n$ HRE from its genomic locus using gRNAs flanking the HRE. To thoroughly evaluate the efficiency of gene editing and the ability to confer gene editing in vivo, we employed both human and murine-derived C9ORF72-related model systems. We utilized patient-derived iPSCs, induced motor neurons (iMNs) and brain organoids with the endogenous mutation as well as primary neurons with expanded repeats from three different C9BAC transgenic mouse models. Furthermore, we tested our gene-editing approach in vivo in three different C9BAC transgenic mouse models and show that HRE excision results in a dramatic reduction of RNA foci and toxic poly-dipeptides—both gain-of-function pathologies caused by the HRE—while rescuing transcript and protein levels.

## Results

### Screening gRNAs flanking the *C9ORF72* HRE in HEK293T cells

To excise the HRE from *C9ORF72*, we designed gRNAs flanking the repeat sequence. Since the HRE is located within an intronic region, our aim was to excise the repeat without affecting exon sequences. However, only 35 base pairs (bp) separate the HRE from exon 2, complicating our strategy of targeting intronic sequence flanking the HRE. Therefore, as an alternative strategy, we used gRNAs targeting intron 2 to excise the HRE and exon 2, a noncoding exon present only in V2 mRNAs. An assumption of this strategy is that excision of the noncoding exon 2 does not affect overall *C9ORF72* mRNA and protein levels, whether single or bi-allelic editing occurs. Four gRNAs were designed flanking the HRE and exon 2, with two gRNAs targeting the 3' DNA sequence downstream of the HRE and two targeting the upstream, 5' sequence (Fig. 1a).

To evaluate gRNA efficiencies, HEK293T cells were co-transfected with a plasmid expressing *S. pyogenes* Cas9 (SpCas9) and another plasmid expressing two gRNA combinations flanking the repeat in groups of gRNAs 1 and 3, gRNAs 1 and 4, gRNAs 2 and 3, or gRNAs 2 and 4, respectively. Cell lysates were collected 48 h after transfection, genomic DNA was isolated, and the targeted region amplified by PCR with primers flanking the HRE sequence (Fig. 1a, b). In HEK293T cells, *C9ORF72* contains only three $G_4C_2$ repeats. Given the lower GC content from this small number of repeats, DNA polymerase is able to amplify the sequence yielding a 523 bp band in unedited *C9ORF72*. Excision of

the target sequence and subsequent repair of the DNA template produces a shorter amplicon of 249 bps for gRNA1,3, 246 bps for gRNA1,4, 322 bps for gRNA2,3, and 321 bps for gRNA2,4. Cells transduced with a plasmid expressing only GFP and untreated cells were used as experimental controls, producing only one amplicon of 523 bps, confirming the presence of unedited template sequences in non-SpCas9 and gRNA-treated cells (Fig. 1b). In contrast, *C9ORF72* editing was observed for all four combinations of gRNAs, as indicated by the presence of smaller amplicons resolved by electrophoresis. Genome editing was very efficient using gRNA2,3 and gRNA2,4 as indicated by the presence of bright bands at 322 bp and 321 bp, respectively, with little amplification of 523 bp products that are produced from unedited template DNA.

We next estimated the percentage of segmental deletions after editing of HEK293T cells using Sanger-sequencing, to test which gRNA pairs were most effective. With gRNA2,3 and gRNA2,4, we achieved over 80% editing efficiency (Fig. 1c). Insertion and deletion (InDel) analysis for gRNA2,3-mediated genome editing demonstrated 65.4% of correct editing and 23.2% excision of the HRE plus insertion of 1 nucleotide (Fig. 1d). For gRNA2,4, 22.6% correct editing was achieved, with the editing of the HRE plus insertion of 1 or 2 nucleotides in 66% of editing events (Fig. 1d). Since these small insertions are located within the intron, they do not change amino acid composition of the C9ORF72 protein and are unlikely to change transcription levels. Since editing efficiencies with Sanger-sequencing are only estimates because of amplification bias and to further investigate on-target genomic rearrangements after CRISPR–Cas9 editing, we assessed the rate of intended HRE deletions or other types of genomic rearrangements with the UDiTaS unidirectional sequencing approach[43], using a single locus-specific primer and a transposon specific primer containing unique molecular identifiers for library construction. UDiTaS analysis at the gRNA2,3 and gRNA2,4-edited loci revealed primarily excision of the HRE. As observed by the amplicon deep sequencing, all groups, including control groups, demonstrated a 2.6–4.9% InDel rate. Additional analysis showed these mostly consisted of 6 bp deletions, which was likely due to the HEK293T cells containing a different number of hexanucleotide repeats than the hg38 assembly we used for analysis. Inversions accounted for <0.4% and were not significantly higher than in the control groups and there was no evidence of further on-target genomic rearrangements. The SpCas9 endonucleases can generate unintended editing at off-target sites throughout the genome[44], particularly at sites with sequence homology to gRNAs. Since unintended editing is a potential safety concern, we performed an in silico analyses of the human genome (Supplementary Table 1) to identify off-target sequences with less than three mismatches with gRNA target sequences. For coding sequences, we identified 2 mismatch targets for gRNA2 and gRNA3 and three mismatch targets for gRNA4. Using amplicon deep sequencing, we interrogated genomic DNA from edited HEK293 cells, but found no evidence of off-target editing at near-cognate sequences (Supplementary Fig. 1), indicating high guide-specificity of genome editing. Therefore, combinations gRNA2,3, and gRNA2,4, and SpCas9 were packaged in separate AAV9 vectors and used in all subsequent in vitro and in vivo experiments.

### Excision of the HRE in vitro

While we achieved highly efficient editing in HEK293T cells, they have only 3 $G_4C_2$ repeats. Therefore, we next tested the editing system in mouse primary cortical neurons from a *C9ORF72* BAC111 transgenic line with 600 repeats, a more disease-relevant in vitro model system[45]. BAC111 mice were crossed with B6J.129(Cg)-*Gt(Rosa)26Sor*$^{tm1.1(CAG-cas9*,-EGFP)Fezh}$/J to create *het:het* C9$^+$/Cas9$^+$ embryos where Cas9 is endogenously expressed in neurons harboring the HRE expansion. Primary cortical neurons were then harvested at

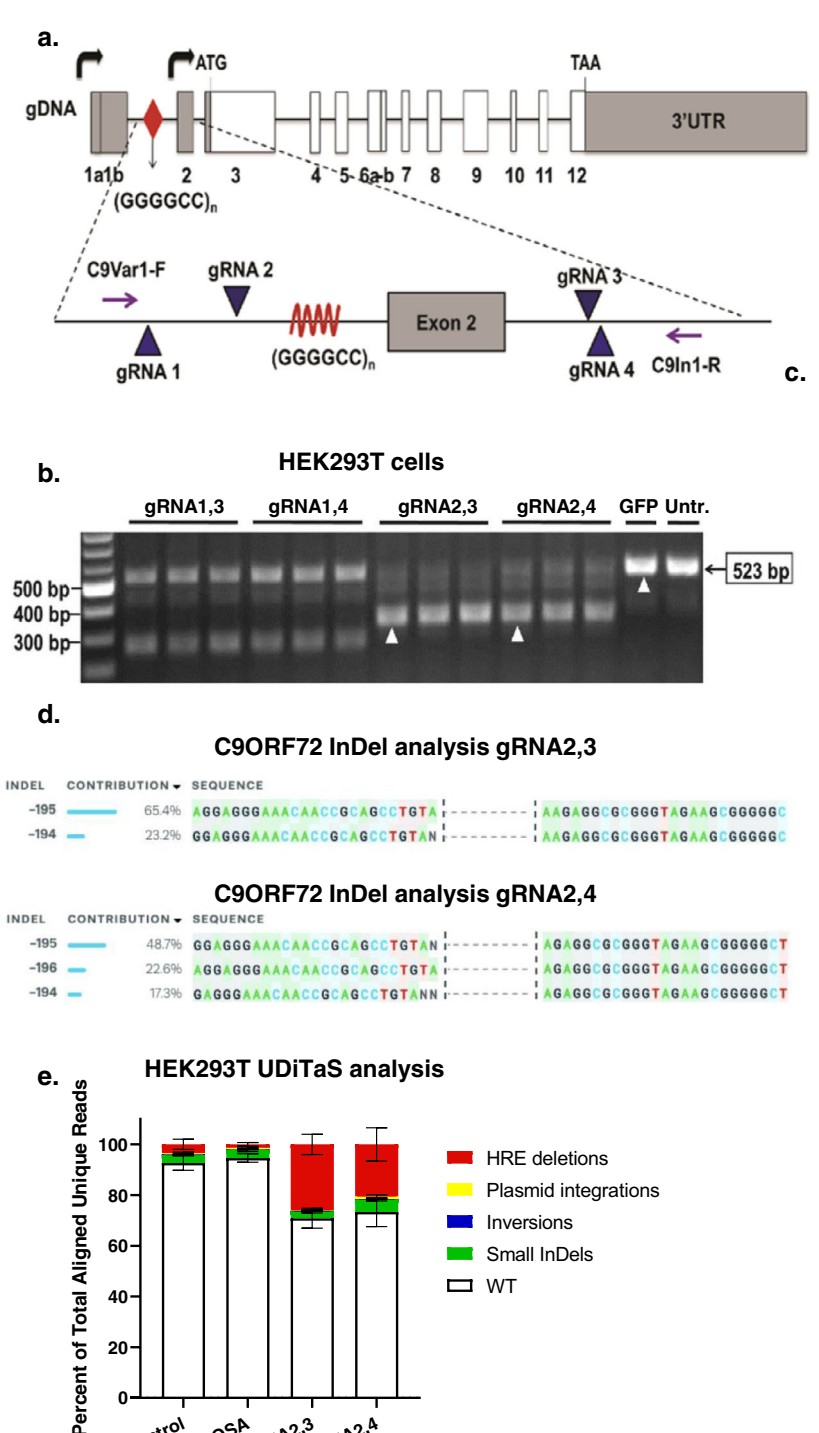

**Fig. 1 | Screening of gRNAs flanking the (GGGGCC)ₙ HRE for excision, in HEK293T cells. a** Human *C9ORF72* gene showing (GGGGCC)ₙ repeats (red diamond) and the locations and sequences of guide RNAs 1, 2, 3, and 4 in addition to primers C9Var1-F and C9In1-R in relation to exon 2 and the (GGGGCC)ₙ repeat. **b** Screening of gRNAs in HEK293T cells. The CRISPR-targeted *C9ORF72* HRE locus was PCR-amplified by C9Var1-F and C9In1-R primers and PCR products were visualized by agarose gel electrophoresis. The expected unedited PCR product size is 523 bp; edited PCR product size for 1–3 and 1–4 is ~250 bp; edited PCR product size for 2,3 and 2,4 is ~320 bp. **c** Percentage of segmental deletions of the HRE in gRNA1–3, 1–4, 2,3, 2,4-treated HEK293T cells measured after Sanger-sequencing. Mean ± SEM, *n* = 3 biological independent samples per group. **d** InDel analysis of gRNA2,3 and 2,4 after Sanger-sequencing, using ICE. **e** Composition of edited alleles at gRNA2,3 and gRNA2,4 editing loci in C9ORF72 in HEK293T cells by UDiTaS analysis. The graphs show the percentage HRE deletions, InDels, plasmid insertions, and inversions. Mean ± SEM, *n* = 4 biological-independent samples per group.

embryonic day 15 (E15) and transduced with AAV9 expressing gRNA2,3 or gRNA2,4 at the day in vitro (DIV) 4 at a multiplicity of infection (MOI) of 50,000 (Fig. 2a). For these experiments, a gRNA targeting the *Rosa* locus was packaged in AAV9 and used as a control since this sequence is not naturally expressed in mice or humans. Since these motor neurons express *C9ORF72* with an expanded repeat (600 repeats), the Q5 DNA polymerase used for PCR is not expected to amplify through this GC-rich region. Therefore, PCR

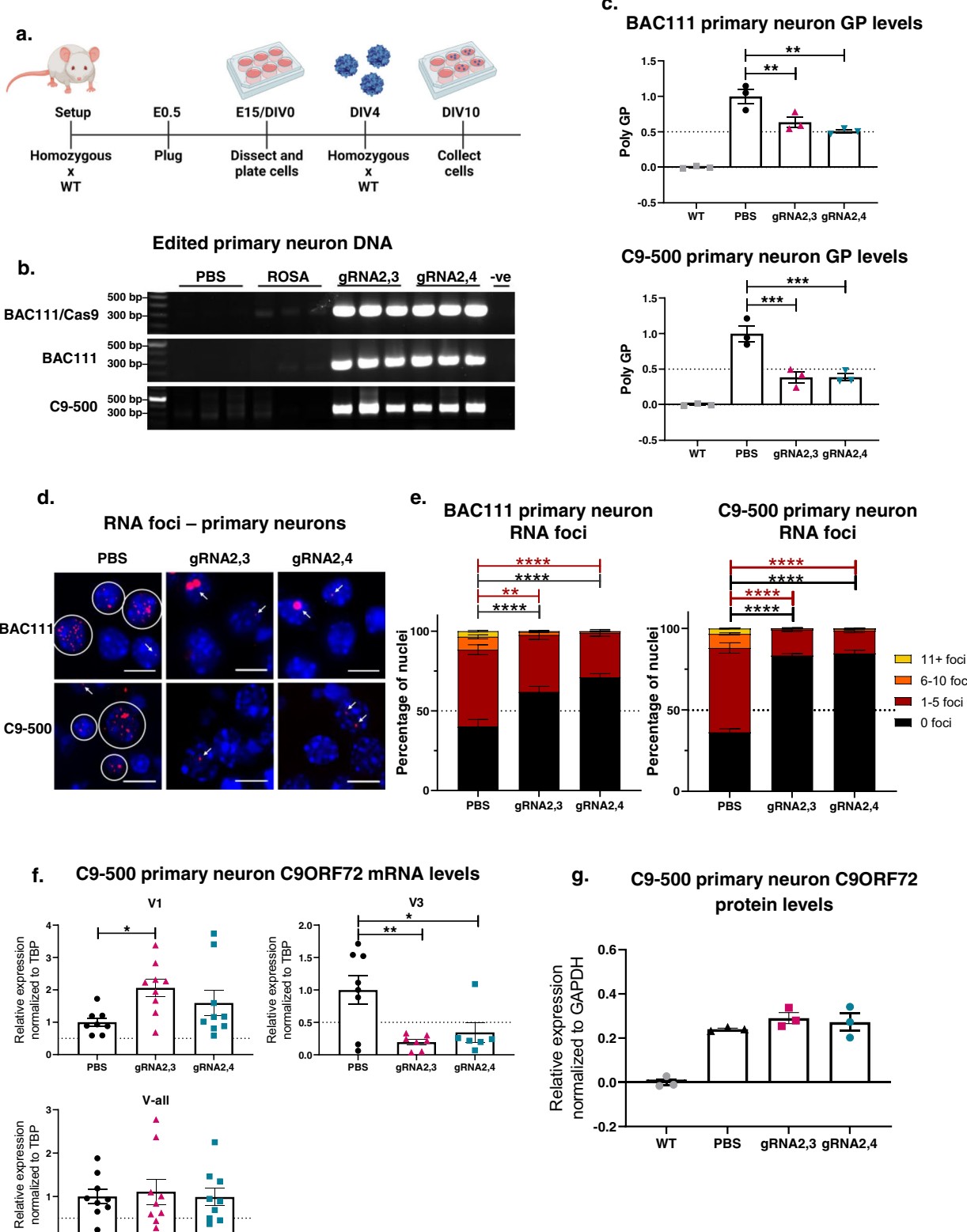

amplification with primers flanking the HRE will only occur if the $G_4C_2$ repeats are excised during gene editing. Following PCR, we observed no DNA amplification in untreated primary neurons or those treated with AAV9-Rosa. Alternatively, amplification of template DNA isolated from neurons transduced with AAV9-gRNA2,3 and AAV9-gRNA2,4 produced a strong band at around 320 bps, demonstrating efficient excision of the HRE mutation in neurons (Fig. 2b).

A limitation of this experiment is that we used neurons that endogenously express Cas9. While this is more efficient, it is also less clinically relevant because Cas9 does not need to be introduced exogenously using AAV vectors. Therefore, to assess editing efficiency in non-Cas9-expressing primary neurons, we used AAV vectors to introduce Cas9 and gRNA expression in primary neurons from heterozygous BAC111 C9+/Cas9- embryos. As in neurons with endogenous Cas9 expression, we observed efficient excision of the HRE mutation in

**Fig. 2 | In vitro gene editing of embryonic primary cortical neurons results in a decrease of pathological hallmarks in C9ORF72 ALS/FTD. a** Schematic of harvest and AAV9-delivered Cas9 and gRNA treatment of different C9ORF72 BAC-transgenic mouse cortical primary neurons. Homozygous C9ORF72 BAC-transgenic mice were crossed with WT mice, resulting in heterozygous mice. To generate BAC111/Cas9 mice, homozygous BAC111 mice were crossed with homozygous B6J.129(Cg)-*Gt(Rosa)26Sor*^tm1.1(CAG-cas9*,-EGFP)Fezh^/J resulting in heterozygosity of both BAC111 C9ORF72 and Cas9. Cells were harvested at embryonic day (E) 15 and transduced with AAV9-gRNA2,3, AAV9-gRNA2,4, and controls at the day in vitro (DIV) 4 and collected at DIV10 for further analyses. BAC111 and C9-500 primary neurons were treated with both AAV9–SpCas9 and AAV9-gRNA2,3 or AAV9-gRNA2,4 and BAC111/Cas9 with only AAV9-gRNA2,3 or AAV9-gRNA2,4. **b** Genomic DNA of treated BAC111/Cas9, BAC111, and C9-500 primary neurons was amplified with primers flanking gRNA seed sequences, and PCR products were resolved by electrophoresis. The expected band for gRNA2,3 and gRNA2,4-edited DNA is 680 bps. No band is expected for unedited DNA. **c** Relative expression of poly-GP in BAC111 and C9-500 primary neurons treated with AAV9-Cas9, and AAV9-gRNA2,3 or AAV9-gRNA2,4, assayed by sandwich immunoassay. Data represent mean ± SEM,

$n = 3$ per experimental group, one-way ANOVA, Dunnett, **$P = 0.0093$ and $0.0016$, ***$P = 0.0007$ and $0.0007$. **d** Representative images of sense RNA foci visualized by fluorescent in situ hybridization (FISH) in BAC111/Cas9 and C9-500 primary neurons treated with AAV9-gRNA2,3 and AAV9-gRNA2,4 for 6 days. Scale bars represent 10 µm. **e** Quantification of sense foci in treated and untreated BAC111/Cas9 and C9-500 primary neurons. The bar graph represents average percent of nuclei containing 0, 1–5, 6–10, and 11+ foci. Three independent experiments and >600 nuclei counts per sample. Two-way ANOVA, Tukey: **$P = 0.0018$, ****$P < 0.0001$. **f** ddPCR analysis of mRNA expression levels of variants V1, V3, and all variants jointly (V-all) after AAV9-gRNA2,3 and AAV9-gRNA2,4 treatment in C9-500 primary neurons. Expression levels were normalized to TBP and compared to the PBS-treated control. Mean ± SEM, PBS $n = 8$, treated groups $n = 9$, one-way ANOVA, Dunnett, *$P = 0.0323$ and $0.0238$, **$P = 0.0044$. **g** Densitometric quantification of the C9ORF72 long isoform in C9-500 primary neurons treated with AAV9-gRNA2,3 and AAV9-gRNA2,4. C9 protein levels were normalized against total protein levels. Levels from primary neurons from wild-type mice were averaged and subtracted as a background signal. Mean ± SEM, $n = 3$, one-way ANOVA, Dunnett, differences between PBS and AAV9-gRNA2,3, AAV9-gRNA2,4-treated cells were not significant.

neurons transduced with AAV–Cas9 and either AAV9-gRNA2,3 or AAV9-gRNA2,4, but not in neurons treated with buffer (PBS) or AAV9-Rosa (Fig. 2b). The BAC111 mouse model harbors 6–8 copies of exon 1–6 of the human *C9ORF72* gene, including 600 G₄C₂ repeats, and recapitulates the molecular pathology of C9-ALS/FTD since mice develop RNA foci and toxic dipeptides[45]. This model is in contrast with C9-ALS/FTD patients, who have only one copy of the C9ORF72 gene with a pathological expansion[46]. Therefore, we validated our system using another transgenic mouse model (C9-500) developed by Dr. Laura Ranum at the University of Florida[47]. The C9-500 mouse expresses only one complete copy of the human *C9ORF72* gene with 500 G₄C₂ repeats and does not express Cas9 endogenously. Using the same experimental approach as before, we transduced primary neurons from C9-500 transgenic mice with AAV9–SpCas9 and AAV9-gRNA2,3 or AAV9-gRNA2,4. Again, we observed efficient excision of the HRE as indicated by efficient amplification of the *C9ORF72* target sequence (Fig. 2b).

## Pathological hallmarks are decreased by excision of the HRE in vitro

Having observed efficient removal of the HRE, we next sought to determine whether genome editing ameliorates two pathogenic gain-of-function hallmarks of C9-ALS/FTD: RNA foci and production of toxic poly-dipeptide proteins. To this end, primary neurons from BAC111 and C9-500 transgenic mice were treated with AAV9-gRNA2,3 or AAV9-gRNA2,4 and AAV9–SpCas9 (Fig. 2a). To evaluate poly-dipeptide production, we used a sensitive immunoassay[48] to quantify levels of GP, an abundant poly-dipeptide synthesized from expanded sense and antisense *C9ORF72* transcripts. Both gRNA combinations resulted in a significant ~50% reduction of GP poly-dipeptides in both neuronal models (Fig. 2c). Next, we labeled RNA foci by fluorescent in situ hybridization (FISH) staining (Fig. 2d) and binned nuclei according to the percentage of observed foci in BAC111 and C9-500 primary neurons (1–5 foci, 6–10 foci, and 11+ foci). Six days after AAV9-gRNA2,3 and AAV9-gRNA2,4 treatment, the percentage of nuclei with foci was markedly decreased, from 64% in the untreated PBS condition to 17% (AAV9-gRNA2,3) and 16% (AAV9-gRNA2,4) in the treated conditions in C9-500 primary neurons (Fig. 2e). Similarly, in BAC111 primary neurons, the percentage of nuclei with foci decreased after treatment from 60 to 38% (AAV9-gRNA2,3) and 29% (AAV9-gRNA2,4). Curiously, editing resulted consistently in enlarged foci in BAC111 primary neurons, but not C9-500 primary neurons (Fig. 2d). To exclude the possibility that the formation of large foci is caused by the accumulation of excised DNA, slides with primary neurons were treated with DNase or RNase prior to analysis by FISH. We found that DNase resulted in degradation of genomic DNA but labeled HRE foci remained. In contrast,

treatment with RNase degraded all foci, including large foci. These findings indicate that excised HRE DNAs do not contribute to the formation of HRE foci and that enlarged foci observed in BAC111 primary neurons is a consequence their having more copies of the expanded *C9ORF72* transgene relative to C9-500 neurons. Collectively, these results demonstrate that both gRNA pairs acting with CRISPR/Cas9 nuclease correctly excised the HRE and thereby reduced levels of RNA foci and poly-dipeptide proteins in vitro; that is, they are both effective in reducing gain-of-function effects in neurons containing the human C9ORF72 HRE.

One proposed mechanism by which the HRE leads to pathology is through reduced expression of C9ORF72 protein through epigenetic repression[49–51], inhibition of RNA processing[12,52,53], or changes in the relative expression of RNA variants[2]. Through alternative splicing and transcription start site utilization, three protein-coding *C9ORF72* transcripts are produced, encoding two isoforms of the C9ORF72 protein[2]. Transcript variants 2 (NM_018325.3) and 3 (NM_001256054.1) encode a 481 amino acid protein, whereas variant 1 (NM_145005.5) encodes a shorter 222 amino acid protein[2]. The V2 variant constitutes the majority (~80%) of *C9ORF72* transcripts in the brain, whereas V1 and V3 represent 20% and less than 3%, respectively[54]. Given the evidence that *C9ORF72* haploinsufficiency may contribute to pathology, we intentionally designed gRNA pairs to exclusively target non-protein-coding DNA. In addition, to determine whether gene editing restores normal expression levels of *C9ORF72* transcripts, we used digital droplet PCR (ddPCR). Given the location of gRNA2,3 and gRNA2,4, we were unable to design ddPCR probes to specifically target V2; however, we could indirectly infer V2 levels using probes for V1, V3 and a probe that amplifies all three variants. We extracted RNA from treated and untreated C9-500 primary neurons and normalized expression levels to TATA-box Binding Protein (TBP). Gene editing increased V1 transcription while a significant decrease in V3 expression was observed (Fig. 2f). Since V3 expression makes up for less than 3% of all C9 transcripts[54], there appeared to be little effect on total transcriptional output from the locus with no significant effect observed using a probe that detects all three transcripts (Fig. 2f). Likewise, levels of the larger, dominantly expressed C9ORF72 protein isoform were unchanged in edited C9-500 primary neurons relative to unedited neurons (Fig. 2g). Together, these results demonstrate that excision of the HRE does not significantly impact C9ORF72 transcription or protein levels, a consequence that could have deleterious effects.

## RNA foci and poly-dipeptides are decreased by excision of the HRE in vivo

Having established the editing efficacy using AAV9-gRNA2,3 and AAV9-gRNA2,4 in different C9ORF72 primary neurons from multiple transgenic mouse models, we next evaluated their efficacy in vivo using

three transgenic mouse models: BAC111/Cas9, BAC112, and C9-500 mice. Since BAC111/Cas9 mice express Cas9 endogenously, the performance of gRNAs can be evaluated more effectively. Since BAC112 mice have multiple copies of the expanded C9ORF72 gene but do not express Cas9, we were able to evaluate the efficiency of gene editing using a dual vector system. Since the C9-500 mouse has only one copy of the HRE expansion, it is more clinically relevant - from a gene-editing perspective. While none of these mouse models present the progressive neurodegeneration seen in ALS and FTD patients, they all exhibit gain-of-function pathological features seen in patients, such as RNA foci, and RAN translation resulting in the presence of toxic poly-dipeptides. Young adult BAC111 mice (aged 2–3 months) were injected bilaterally in the striatum with AAV9-gRNA2,3 or AAV9-gRNA2,4, ($7 \times 10^9$ vector genomes (VG) of the gRNA vectors) while BAC112 and C9-500 mice additionally received $2 \times 10^{10}$ VG of AAV9-Cas9. Eight weeks following injection, mice were sacrificed, and striatal tissues from both brain hemispheres were harvested (Fig. 3a). To evaluate gene-editing events, we again used a PCR assay with primers flanking the HRE sequence, whereby amplification only occurs in the presence of edited DNA but not for templates with the long GC-rich HRE sequence (BAC111: 600 repeats, BAC112: 550 repeats, C9-500: 500 repeats). As observed in primary neurons, gRNA2,3 and gRNA2,4-edited DNA produced the expected amplicon, while equivalent amounts of DNA from PBS and AAV9-Rosa/AAV9-Cas9 treated animals did not (Fig. 3b). These results document successful in vivo HRE excision in three different transgenic mouse models of *C9ORF72* ALS and FTD.

Having observed excision of the HRE, we next sought to determine whether this attenuates toxic gain-of-function pathologies in vivo. As before, we used ELISA and FISH to quantify poly-dipeptide and HRE RNA foci levels, respectively. We found levels of poly-GP[24], to be significantly decreased in the striatum of C9-500 mice infused with AAV9–SpCas9 and either AAV9-gRNA2,3 or AAV9-gRNA2,4, but not in mice treated with PBS (Fig. 3c). Although poly-GP is abundant and commonly used as a biomarker for overall poly-dipeptide production, it is less toxic than other poly-dipeptides. The arginine-containing poly-dipeptides, poly-GR, and poly-PR, are particularly cytotoxic[28]. The currently-identified mechanisms contributing to poly-GR toxicity include impaired protein translation[55,56], formation of stress granules[29], inhibition of DNA repair[57] and reduced mitochondrial functionality[37]. To validate that HRE excision reduces levels of arginine-containing poly-dipeptides, we evaluated the effects of HRE excision on poly-GR levels. We found a significant and robust reduction of poly-GR levels after treatment with AAV9-gRNA2,3 and gRNA2,4 in the striatum of BAC111/Cas9 (50% reduction) and in C9-500 mice (60% reduction) (Fig. 3d).

Next, we quantified the percentage of nuclei with HRE RNA foci and assigned nuclei to one of several bins: 1–5 foci, 6–10 foci and 11+ foci. For C9-500 mice, the number of nuclei in the striatum with foci was significantly and robustly reduced from 86% in the untreated condition to 20–33% in the treated conditions. As in the BAC111/Cas9 mice, the number of striatal nuclei with foci was reduced from 44% in controls to 17–18% in the treated animals. Moreover, fewer foci per cell were detected in treated animals, where both gRNA combinations are potent inhibitors of $G_4C_2$ RNA foci (Fig. 3e, f). As in primary neurons, V-all transcript levels and C9ORF72 long isoform protein levels remained unchanged after treatment. Collectively, these results strongly suggest that removal of the HRE corrects toxic gain-of-function pathologies (RNA foci and poly-dipeptides) in vivo and are unlikely to induce further haploinsufficiency (Fig. 3g, h).

## Assessment of editing outcomes in vivo

As proof-of-mechanism, we delivered Cas9 and gRNAs by AAV. Because of the large size of SpCas9 (4.1 kb) we delivered SpCas9 and the gRNA pairs separately in two AAV9 vectors in C9-500 mice. Since gene editing will only occur when a cell is transduced with both the nuclease and gRNAs, we stained both using RNAScope (Fig. 4a) and counted how many cells were transduced by both vectors after striatal injections in C9-500 mice. We found that most cells in the striatum were transduced by both AAVs, 58% for gRNA2,3 and 69% for gRNA2,4-treated C9-500 mice (Fig. 4b).

To investigate editing efficiencies after treatment with SpCas9 and gRNA2,3 or gRNA2,4 in C9-500 mice, we first used PacBio No-Amp targeted sequencing to sequence through the editing sites and the GC-rich HRE[58]. After two runs with limited numbers of on-target reads of mouse striatal DNA samples, we opted to use UDiTaS to measure HRE excision and investigate on-target rearrangements and AAV integrations. UDiTaS analysis at the gRNA2,3 and gRNA2,4-edited loci revealed 15% and 23% HRE excision, respectively, without large InDels (Fig. 4d). As to be expected[59], we found high levels of AAV integration at the gRNA2,3 and gRNA2,4 editing site (Fig. 4d). Since UDiTaS captures 140 bp from each end of the amplicons, we could not distinguish between AAV integrations paired with removal of the HRE, and AAV integrations in which the HRE remained intact. To resolve this, we resorted to analyzing the PacBio No-Amp data on similarly treated samples and determined that 78.6% of AAV integrations were paired with the removal of the HRE. This level of HRE excision supports the previous data in which we found a >50% (Fig. 3c–f) reduction of RNA foci and RAN translation, and >50% of striatal cells were transduced with Cas9 and gRNAs in treated C9-500 mice (Fig. 4b). We found that gRNA2,3 and gRNA2,4-treated samples showed a small percentage of InDels (3.1% and 1.6%), although small indels within the noncoding intron would not be expected to have deleterious effects, however, our analyses do not detect the occurrence of large deletions[60] if such deletions involve the single locus-specific UDiTaS primer binding site. Interestingly, the percentage of inversions was higher in treated C9-500 brains, than in HEK293T cells (Fig. 1e); however, due to the relatively short UDiTaS read length we were not able to see if these inversions still had the HRE or if only the 3' end of the excised sequence was inversed and back-integrated.

## All disease mechanisms are rescued by excision of the HRE in patient cells

To investigate the effects of HRE excision with AAV9-gRNA2,3 and gRNA2,4 on C9ORF72 pathology in human motor neurons, we used an iPSC line from an C9-ALS/FTD patient carrying a pathogenic *C9ORF72* HRE or a control line derived from an individual with fragile x syndrome with a repeat expansion in the *FMR1* gene[61]. Both iPSC lines were derived from lymphocytes and used to produce induced motor neurons (iMNs), as we previously reported[61]. The resulting iMN cultures were co-transduced with AAV9–SpCas9 and either AAV9-gRNA2,3 or AAV9-gRNA2,4. After 7 days, genomic DNA was harvested and analyzed for genome editing using PCR and primers spanning the HRE. Notably, C9-ALS/FTD patient-derived cells harbor one expanded *C9ORF72* allele and one allele with a WT, unexpanded $G_4C_2$ repeat sequence. Transgenic mice used in this study do not have WT human *C9ORF72* alleles,; the transgene harbors only the expanded HRE sequence. Therefore, unlike cells from transgenic mice, in human patient-derived cells, the PCR assay will produce a single amplicon of 523 bps due to amplification of the WT allele template and the absence of amplification through the expanded allele. Healthy individuals, or patients with fragile x syndrome, have two WT *C9ORF72* alleles and will therefore produce the same 523 bp amplicon in the PCR assay. Since the gRNAs 2,3 and 2,4 will induce excision of the $G_4C_2$ repeats within WT alleles as well as excision of the HRE in expanded alleles, the result is a shift from 523 to 320 bps when editing occurs in either patient or control cell lines. We found that co-transduction of C9-ALS/FTD iMNs resulted in a shift from amplification of a 523 bp product to the 320 bp product in cells treated with AAV9–SpCas9 and either gRNA pair, but not in C9-ALS/FTD or fragile X syndrome control iMNs without transduction with AAV9–SpCas9 (Fig. 5a). However, based on this PCR assay

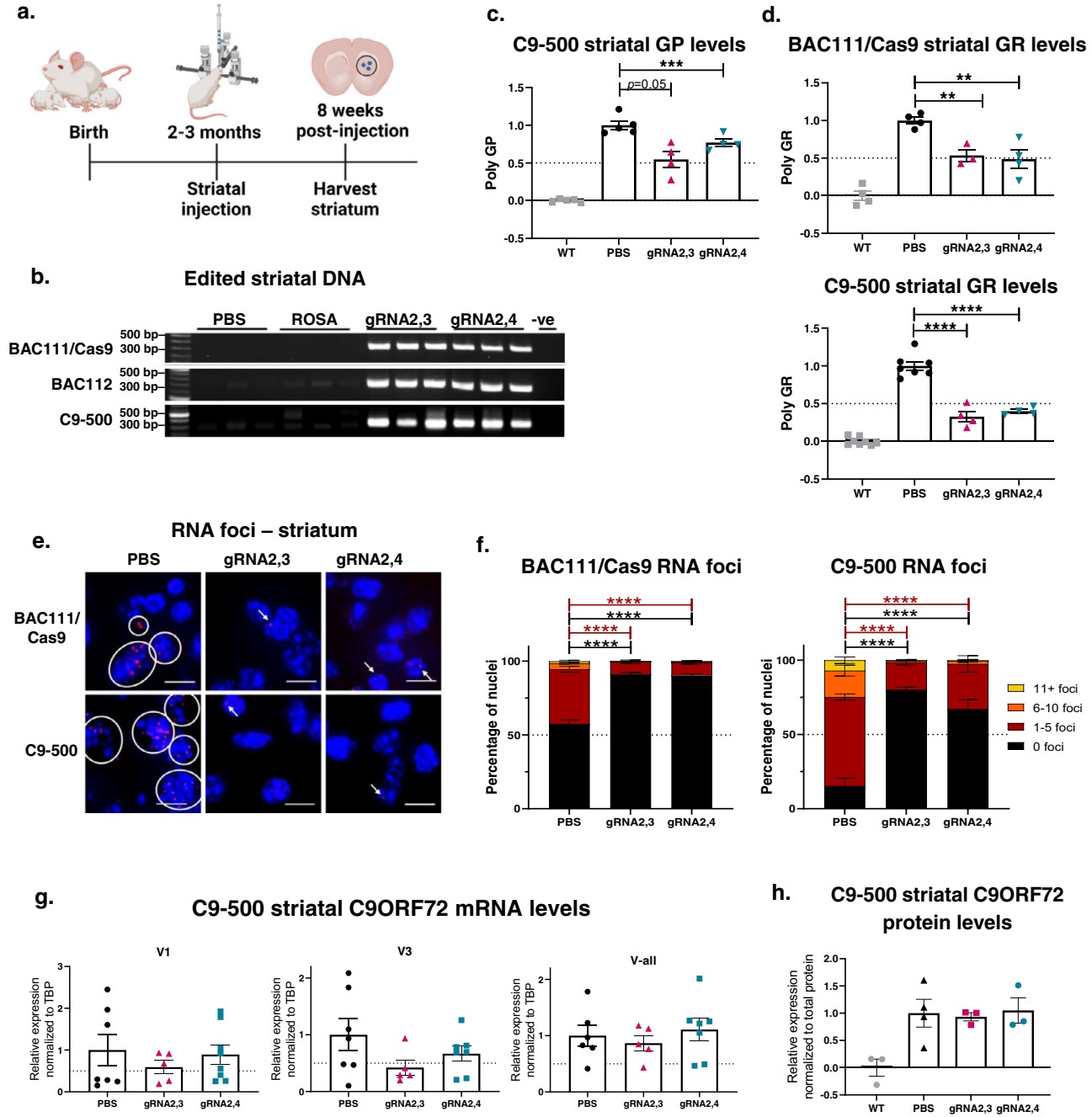

**Fig. 3 | In vivo gene editing results in a decrease of RNA foci and poly-dipeptides in different C9BAC-transgenic mouse models. a** Schematic of treatment and harvest of AAV9-delivered Cas9 and gRNA treatment of different C9ORF72 BAC-transgenic mouse models. Mice were treated at 2–4 months of age with bilateral striatal injections, and the striatums were harvested 8 weeks after treatment for further analyses. BAC111, BAC112, and C9-500 mice were treated with both AAV9–SpCas9 and AAV9-gRNA2,3 or AAV9-gRNA2,4 and BAC111/Cas9 mice with only AAV9-gRNA2,3 or AAV9-gRNA2,4. **b** Genomic DNA of treated BAC111/Cas9, BAC112, and C9-500 mouse striatums was amplified with primers flanking gRNA seed sequences, and PCR products were resolved by electrophoresis. The expected band for gRNA2,3- and gRNA2,4-edited DNA is 680 bps. No band is expected for unedited DNA. AAV9-Rosa treatment and PBS treatment were used as a control. Genomic DNA was amplified with primers flanking gRNA seed sequences, and PCR products resolved by electrophoresis. The expected band for gRNA2,3- and 2,4-edited DNA is 680 bps. No band is expected for unedited DNA. **c** Relative expression of poly-GP in the striatums of treated C9-500 mice. Data represent mean ± SEM, WT and PBS $n = 5$, treated groups $n = 4$, one-way ANOVA, Dunnett, $P = 0.0502$, ***$P = 0.0004$. **d** Relative expression of poly-GR in the striatums of treated BAC111/ Cas9 and C9-500 mice. Data represent mean ± SEM, BAC111/Cas9 WT, PBS,

gRNA2,4-treated mice $n = 4$, gRNA2,3 treated mice $n = 3$; C9-500 WT $n = 6$, PBS $n = 7$, both gRNA-treated groups $n = 4$, one-way ANOVA, Dunnett, **$P = 0.0085$ and 0.0026, ****$P < 0.0001$. **e** Representative images of sense RNA foci visualized by FISH in BAC111/Cas9 and C9-500 striatums treated with AAV9-gRNA2,3, AAV9-gRNA2,4, and PBS. Scale bars represent 10 μm. **f** Quantification of sense foci in treated BAC111/Cas9 and C9-500 mice striatums. Bar graph represents average percent of nuclei containing 0, 1–5, 6–10, and 11+ foci. $n = 3$ mice per group, and >600 nuclei were counted per sample. Two-way ANOVA, Tukey: ****$P < 0.0001$. **g** ddPCR analysis of mRNA expression levels of variants V1, V3, and all variants jointly (V-all) after gRNA2,3 and gRNA2,4 treatment in C9-500 mice. Expression levels were normalized to TBP and compared to the PBS-treated control. Mean ± SEM, PBS $n = 7$, gRNA2,3 $n = 5$, gRNA2,4 $n = 8$, one-way ANOVA, there were *no* significant differences between treated and untreated groups. **h** Densitometric quantification of the C9ORF72 long isoform in treated C9-500 mice. C9 protein levels were normalized against total protein levels. Levels from wild-type mice were averaged and subtracted as a background signal. Mean ± SEM, WT, gRNA2,3 and gRNA2,4 $n = 3$, PBS $n = 4$, one-way ANOVA, Dunnett, differences between PBS and gRNA2,3-, gRNA2,4-treated cells were not significant.

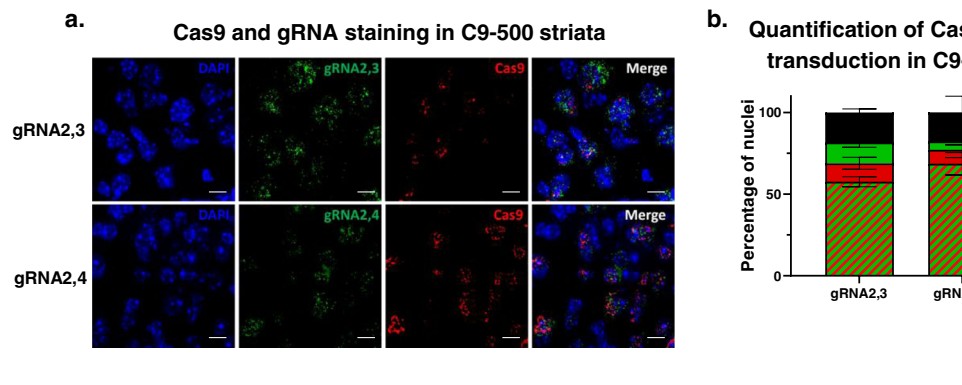

**a. Cas9 and gRNA staining in C9-500 striata**

**b. Quantification of Cas9 and gRNA transduction in C9-500 striata**

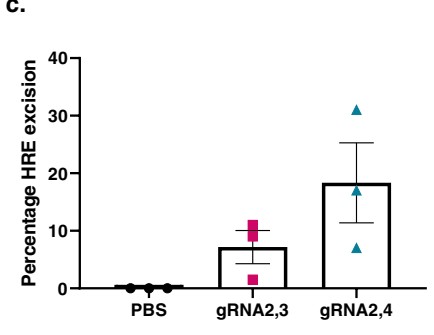

**C9-500 striata – NoAmp PacBio**

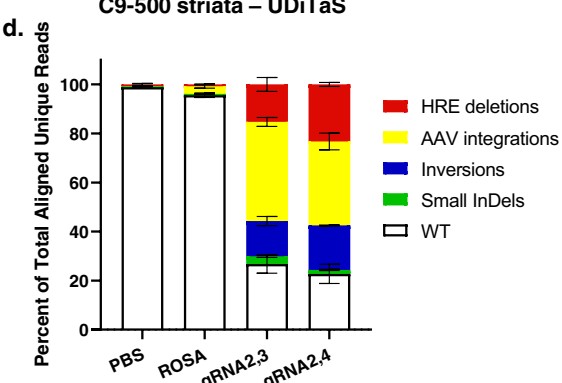

**C9-500 striata – UDiTaS**

**Fig. 4 | In vivo gene-editing quantity and quality in C9BAC-transgenic mouse model. a** Representative ×64 images of C9-500 striata labeled with RNAscope in situ hybridization for Cas9 (red) and GFP (green) expression and nuclei (DAPI, blue) after AAV9-Cas9 and AAV9-gRNA2,3 or AAV9-gRNA2,4 treatment. The gRNA vectors expressed GFP. Scale bars represent 10 μm. **b** Percentage of nuclei stained with Cas9, GFP, Cas9 and GFP, and no staining of AAV9-Cas9 and AAV9-gRNA2,3- or AAV9-gRNA2,4-treated C9-500 mice. **c** Percentage of reads in which the HRE is excised after gRN2,3 and gRNA2,4 editing in C9-500 striatal DNA measured by No-Amp PacBio. Mean ± SEM, $n = 3$ per group. **d** Composition of edited alleles at gRNA2,3 and gRNA2,4 editing loci in C9ORF72 in C9-500 striata by UDiTaS analysis. The graphs show the percentage of HRE deletions, small InDels, AAV insertions, and inversions. Mean ± SEM, $n = 4$ per group.

alone, it is not possible to determine whether gene editing occurs at WT alleles, expanded alleles or both. To address this, iPSC cultures were co-transduced with AAV9–SpCas9 and AAV9-gRNA2,4 and serially plated at low density and single colonies manually selected to obtain clonal iPSC cell populations. Analysis of several clones indicated efficient editing events as indicated by the presence of both 523 and 320 bp bands (Fig. 5b). To ensure clonality of iPSC lines, colonies were subcloned by titration and manual selection. From clone #11, we selected three subclones (4, 6, and 11) to exemplify possible outcomes: no editing, editing of the WT allele alone, or editing of both alleles. Using PCR, clone 11-6 was found to have no editing as indicated by a 523 bp band, while clones 11–4 and 11-11 were found to have editing as indicated by the presence of a 320 bp band (Fig. 5c). To determine whether gene editing of the expanded allele occurred in subclones, we used a repeat-primed PCR to amplify the HRE sequence followed by capillary electrophoresis to resolve amplicons as we have done previously[61]. This analysis confirmed that deletion of the HRE had occurred for subclone 11-11, but not subclone 11–4, as indicated by the presence of a prototypical sawtooth pattern resulting from amplification of the HRE (Fig. 5d). PacBio No-Amp sequencing[58] confirmed 100% excision of the HRE in expanded alleles in subclone 11-11, while the unexpanded WT allele was not edited (Fig. 5e). To assess the quality of the editing, we performed UDiTaS analysis on the subclone 11-11 and we found no genomic rearrangements or AAV integrations at the editing locus (Fig. 5f).

To evaluate the effect of complete HRE excision, we measured levels of GP poly-dipeptides in the parent *C9ORF72* ALS/FTD iPSC line and subclone 11-11. We found a near complete elimination of GP poly-dipeptides in the gRNA2,4-edited 11-11 subclone relative to untreated

parent iPSCs (Fig. 5g). Next, we investigated whether AAV9–SpCas9/gRNA2,4 mediated gene editing could instigate HRE excision in *C9ORF72* ALS/FTD cerebral brain organoids. Cerebral brain organoids are 3D structures cultured from iPSCs that better recapitulate the physical and diverse cell-type composition of the human brain, as compared to 2D iMN cultures. Cultures of *C9ORF72* iPSCs were used to produce brain organoids as described in[62] and transduced with an AAV9-GFP expression vector to visualize transduction (Fig. 5h). Next, organoids were co-transduced with AAV9–SpCas9/gRNA2,4 at DIV 5, DIV17 or both DIV 5 and 17. At DIV 32, organoids were harvested, and genomic DNA was isolated for PCR analysis to detect genome editing events. Successful editing of brain organoids occurred in gRNA2,4-treated organoids after early treatment at DIV 5, late treatment at DIV17, and with treatment at both time points (Fig. 5i). Two rounds of transduction produced the most robust band shift from 523 to 320 bps −indicating more efficient gene editing than was achieved by transduction at the 17-day time point.

Since the HRE is located in the promoter region of V2, the most highly expressed *C9ORF72* variant, one of the potential outcomes of excising the HRE is that reduced C9ORF72 transcription efficiency might exacerbate C9ORF72 haploinsufficiency. Alternatively, since HRE excision decreases the physical distance between V2 and the V1/V3 promoter, HRE excision could potentially increase V2 expression. In the brain of *C9ORF72* BAC-transgenic mice and primary neurons derived from them, *C9ORF72* transcript and protein levels remained largely unchanged after treatment (Figs. 2f, g and 3g, h). However, this could be due to incomplete transduction of tissues, abnormal promoter activity resulting from genomic insertion of the human sequence into the mouse genome, or cross-reactivity of the C9ORF72

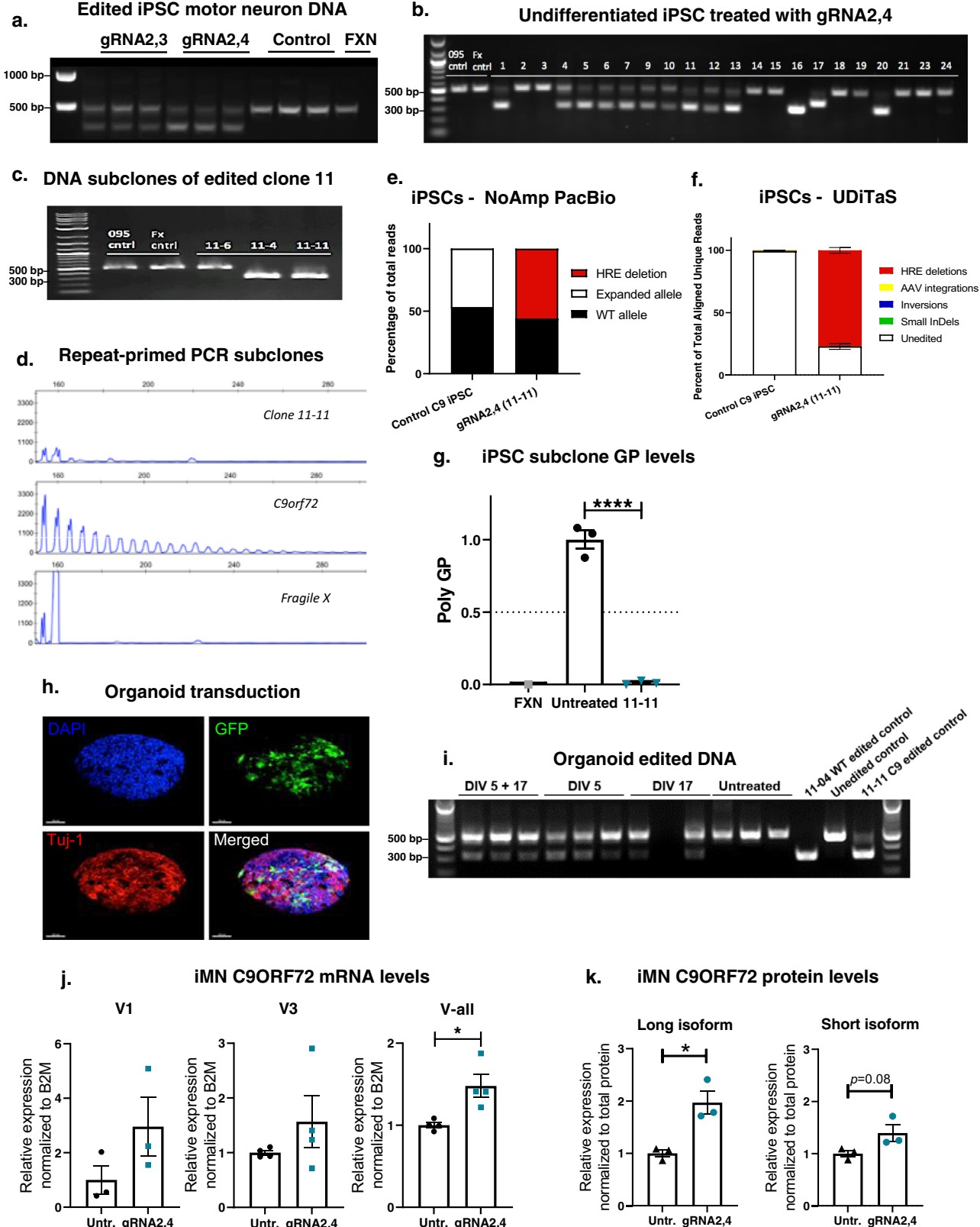

antibody with mouse c9orf72 proteins. Therefore, we took advantage of the iPSC system, in which the HRE exists in its natural genomic location in a human cell. In this model, purified subclones can be compared without the potential confounds of incomplete transduction, inefficient editing or the presence of mouse c9orf72. To determine whether HRE excision alters transcription rates, RNAs and proteins were isolated from parent *C9ORF72* iMNs or the AAV9-

gRNA2,4-edited 11-11 isogenic iMNs. We found a significant increase of almost 50% in *C9ORF72* transcription using primers that detect all three variants and a trend towards increased V1 and V3 levels (Fig. 5j). Western analysis confirmed increased expression of C9ORF72 protein: levels of the long isoform, translated from V2 and V3, were robustly and significantly ($P < 0.05$) increased in gene-corrected iMNs as compared to the parent line (Fig. 5k). Similarly, the short C9ORF72 isoform,

**Fig. 5 | Genome editing of the hexanucleotide repeat ($G_4C_2$)$_x$ expansion in C9ORF72 iPSCs and iMNs. a** C9ORF72 patient iPSC motor neurons transduced with Cas9 and gRNA2,3 and gRNA2,4. Genomic DNA was amplified with primers flanking gRNA seed sequences and PCR products resolved by electrophoresis. DNA isolated from Fragile X syndrome iPSC neurons (FXN) was used as a control. **b** Genome editing of undifferentiated C9ORF72 iPSCs with gRNA2,4. iPSCs were co-transduced with AAV–Cas9 and gRNA2,4, individual colonies (1–24) were manually selected and analyzed for genome editing by PCR using primers spanning the HRE. **c** Clone 11 selected from iPSCs transduced with Cas9 AAV and gRNA2,4 was subcloned to generate lines 11-6, 11–4, and 11-11. Analysis of genome editing revealed editing for subclones 11–4 and 11-11 but not 11-6, as indicted by a shift from 521 to 321 base pair product using primers that span the HRE. **d** Capillary electrophoresis fragment analysis of amplicons produced by repeat-primed PCR of the C9ORF72 HRE for subclone 11-6,11–4, 11-11, and FGX and untreated patient iPSCs. **e** Percentage of reads in which the HRE is excised in gRNA2,4-edited 11-11 isogenic iPSCs measured by No-Amp PacBio. **f** Composition of edited alleles at the gRNA2,4 editing loci in gRNA2,4-edited 11-11 isogenic iPSCs by UDiTaS analysis. The graphs show the percentage of HRE deletions, small InDels, AAV insertions, and inversions. Data represent mean ± SEM, $n = 3$. **g** Relative expression of poly-GP in the striatums of treated C9-500 mice. Data represent mean± SEM, $n = 3$ per experimental group, one-way ANOVA, Dunnett, ****$P < 0.0001$. **h** Representative confocal Z-stack images of organoid transduction. Organoids were stained for DAPI (blue) and immunolabeled with antibodies against Tuj-1 (red) and GFP (green). **i** Genomic DNA was isolated and amplified by PCR using primers spanning the C9ORF72 expansion after transduction by Cas9 and gRNA2,4 AAV vectors at 5 days, 5 and 17 days, 17 days or untransduced. DNA from edited iPSC subclones 11-04, 11-11, and DNA from one unedited line was used as controls. **j** ddPCR analysis of mRNA expression levels of variants V1, V3, and all variants jointly (V-all) in gRNA2,4-edited 11-11 isogenic iMNs. Expression levels were normalized to B2M and compared to untreated control patient iMNs. Mean ± SEM, $n = 3$, two-tailed Student's $t$ test, *$P = 0.0158$. **k** Densitometric quantification of the C9ORF72 long isoform and short isoform in gRNA2,4-edited 11-11 isogenic iMNs. C9 protein levels were normalized against total protein levels. Mean ± SEM, $n = 3$, two-tailed Student's $t$ test, $P = 0.0844$, *$P = 0.0138$.

translated from V1, was increased by 40% (Fig. 5k, $P = 0.08$). Together, these results suggest that excision of the HRE does not exacerbate and indeed rescues haploinsufficiency.

## Discussion

In this study, we successfully excised the ALS/FTD-causing HRE in the first intron of *C9ORF72*, using AAV9 to deliver CRISPR SpCas9 nuclease and two gRNAs flanking the HRE to patient cells, primary neurons and three mouse models. This single intervention attenuated the three major disease mechanisms at play in C9-ALS/FTD. We (1) suppressed the production of poly-GP dipeptides in edited isogenic patient cells and both poly-GP and poly-GR in primary neurons and mice, (2) reduced levels of nuclear sense RNA foci in two *C9ORF72* BAC-transgenic mouse models and primary neurons, and (3) increased of *C9ORF72* mRNA transcript and protein levels by 50% in edited isogenic patient cells, thereby countering HRE-induced haploinsufficiency. Due to technical limitations, we were unable to assess the effects of HRE excision on GP and RNA foci levels on an individual neuron basis.

These findings are consistent with, but extend, other reports demonstrating positive benefit of excising the HRE in C9ORF72. Selvaraj et al.[63] attenuated AMPAR-mediated excitotoxicity in patient isogenic cell lines edited by CRISPR/Cas9 to excise the HRE. The Talbot group used CRISPR/Cas9-mediated HDR to replace the pathogenic expansion in patient cells with a donor template carrying a normal repeat size of ($G_4C_2$)$_2$ to create an isogenic iMN line, and thereby reduced the abundance of RNA foci, poly-dipeptides and methylation levels of *C9orf72*[64]. Both studies elegantly documented that excision of the HRE using CRISPR/Cas9 rescued of components of C9ORF72 pathology, but both employed transfection to edit the patient cells[63,64]. By contrast, we have developed a method for AAV-mediated genome editing in patient-derived iPSC model systems and an AAV-mediated delivery system that we tested in multiple C9BAC transgenic models, including those without endogenous Cas9-expression. Efforts by others using Cas9-expressing mice are less comprehensive and less relevant to developing patient therapies[65]. Moreover, our study confirmed the suppression of poly-dipeptide expression using assays for both poly-GP and poly-GR.

We found that excision of the HRE may rescue *C9ORF72* haploinsufficiency, a widely documented aspect of C9-ALS/FTD pathology[2,49,66–68]. Because there are multiple proposed roles for C9ORF72 in a range of cell types, it has reasonably been proposed that a reduction in C9ORF72 protein levels is deleterious. Such functions for C9ORF72 include: membrane trafficking in autophagy[69], lysosomes[70], and endosomes[69]; immune homeostasis in microglia and macrophages[71,72], and assembly of stress granules[73]. Consistent with these observations, *C9orf72* knockout mice exhibit immune dysfunction and weight loss, but no motor neuron degeneration or impairment[71,72,74], suggesting that haploinsufficiency alone does not cause a C9-ALS/FTD phenotype. Similarly, *C9ORF72* BAC-transgenic mice expressing a transgene with expanded HRE while also expressing endogenous *c9orf72*, do not develop a motor phenotype[45,75,76]. It is striking that in *C9orf72* knockout mice that also express an expanded HRE, gain-of-function pathology is exacerbated[11], resulting in a motor phenotype[11,77]. Despite strong evidence for the role of haploinsufficiency in C9ORF72 pathology, most published therapeutic approaches for C9-ALS/FTD, such as microRNA[54,78] antisense oligonucleotide strategies[12,79], and antibody therapies[80] focus on reducing HRE gain-of-function but not rescue of haploinsufficiency. In this present study, we deliver in vivo gRNAs as well as Cas9, that ameliorates both toxic acquired functions of mutant HRE (RNA foci, poly-dipeptides) and *C9ORF72* haploinsufficiency.

AAV has many advantages and is presently regarded as the safest and most effective gene therapy vector for the CNS, with potentially a long-lived therapeutic effect following a single treatment. Currently, three AAV-based therapies are approved by the FDA for the treatment of Leber congenital amaurosis (Luxturna), and spinal muscular atrophy (SMA) (Zolgensma). Despite this, in this study we employed AAV as merely a delivery tool for a proof-of-mechanism. A non-viral mRNA-mediated approach would be better suited for clinical translation as AAV promotes long-term expression, which is unnecessary for CRISPR/Cas9-mediated editing in the CNS. Furthermore, long-term expression of an active genome editing system with AAV present safety concerns, such as: (1) increased genomic instability and off-target editing due to the sustained activity of an active nuclease[81]; (2) immune responses triggered by long-term expression of a bacterial protein (Cas9)[82]; (3) high AAV genome integration levels at the CRISPR/Cas9-induced double-strand break[59], which we also detected in our in vivo experiments (Fig. 4d). Although some of these issues can be addressed by using self-inactivating AAV vectors[83] or tuneable AAV vectors[84], self-integration of AAV genome fragments remains a concern. Delivery of Cas9 and gRNA mRNA by lipid nanoparticles is currently tested in a clinical trial for the treatment of transthyretin amyloidosis. Encouraging preliminary data shows high levels of on-target editing and no serious adverse events[85]. Safety profiles, delivery efficiencies, and rapid advancements in lipid nanoparticle development make it a preferred choice to deliver Cas9 and gRNAs to excise the HRE in C9ORF72.

CRISPR technology has expanded the reach of gene editing; formerly only a versatile research tool, this technology is becoming a powerful method for the treatment of human genetic disease. New versions of Cas9 and Cas9-like nucleases are much smaller than traditional spCas9, improving delivery and application[86]. Here, we have demonstrated that gRNA2,3 and gRNA2,4 in combination with Cas9 are capable of excising the ALS/FTD-causing HRE in *C9ORF72* in three mouse models, primary neurons, and patient cells. This excision attenuated the three major C9-ALS/FTD adverse, pathologic elements ascribed to the expanded repeats in *C9ORF72*: toxic poly-dipeptides,

RNA foci, and haploinsufficiency. These proof-of-concept findings support the view that CRISPR methodologies should be explored as a strategy to halt or slow disease progression in ALS, FTD and ALS/FTD patients, and potentially other disorders caused by expanded domains of repeated DNA.

## Methods

### Ethical approvals

Induced pluripotent stem cells were created using ALS/FTD patient lymphocytes carrying an expansion of 450 repeats in C9ORF72 collected by Dr. Michael Benatar MBChB, DPhil under an IRB-approved protocol at the University of Miami (The University of Miami Institutional Review Board). The Federal wide Assurance number was FWA00002247, and the Medical Sciences IRB-A registration number was IRB00005621. Patient consent, including express consent permitting transfer and future use of cells in research, was obtained by Dr. Benatar (PI) and approved by the University of Miami Miller School of Medicine Department of Neurology Clinical Research in ALS (Study #20101021, Clinical Research in Amyotrophic Lateral Sclerosis (CRiALS)). Specimens were de-identified to conceal the personal health information of the donor and transferred to the Zeier Laboratory, where cellular reprogramming was carried out as previously described (PMID: 26099177). The Human Subject Research Office and Institutional Review Board at the University of Miami reviewed and provided approval exempting the use of these cells by Dr. Zeier from human subject research oversight on March 12, 2013. These iPSCs were used to create the induced motor neurons and the organoids used in this study.

All mouse experiments were conducted at UMass Medical School following protocols approved by the Institutional Review Board. The University of Massachusetts Medical School Institutional Animal Care and Use Committee approved all experiments involving animals.

### gRNA design and cloning

The CRISPRseek Bioconductor package[87] was used to identify CRISPR/Cas9 target sites flanking the HRE in C9orf72 intron 1 and intron 2 (Fig. 1a). Two gRNAs from each intron with low predicted off-target activity[88] were selected for further analysis. To confirm high activity at these target sites, guide sequences were cloned into the pX330 sgRNA expression plasmid[89] (Addgene plasmid 42230), and the full target site was cloned into pM427, a GFP reporter plasmid for target site cleavage[90]. High nuclease activity was confirmed by induction of GFP-positive cells following co-transfection of these plasmids into HEK293T cells. Next, the guides were cloned in pairs into the 3'end of a CB-GFP encoding AAV proviral plasmid, each under the control of a U6 promoter. Two plasmids pAAV-CB-GFP-C9gR-2,3 and pAAV-CB-GFP-C9gR-2,4 (most efficient at editing HEK293T cells) were packaged into rAAV9 vectors. AAV9-pU1a-SpCas9-RBGpA vector was used to deliver and express SpCas9 (Addgene, 121507). To measure InDels in C9ORF72 after editing in HEK293T cells, cells were harvested 72 h after transfection, and genomic DNA was extracted (Gentra Puregene Tissue Kit, Qiagen). The C9ORF72 locus was amplified by PCR, Sanger-sequenced (Genewiz), and analyzed by the ICE web-based interface (Synthego). InDels up to 20 bp were permitted. To measure off-target effects in predicted targets, loci were amplified by PCR and sequenced by CRISPR Amplicon Sequencing (MGH, USA). Next, paired-end amplicon sequencing datasets of potential off-target editing sites were analyzed using CRISPResso2[91]. FASTQ files were trimmed for 3' adapters and low-quality base calls were analyzed in Batch mode with r1 and r2 inputs. "Guide" parameter (-g) specified the sgRNA binding site projected by the in silico analysis, not the near-cognate on-target guide sequence. Cleavage offset was set to −3. Editing quantification ignored 15 bp from read ends, ignoring substitutions (--ignore_substitutions) to exclude erroneous base calls. The AMACR locus was analyzed using 150 bp, single-end reads (-r1 flag only) in the manner described above.

### Animal experiments

All mouse experiments were conducted at UMass Medical School following protocols approved by the Institutional Review Board. The University of Massachusetts Medical School Institutional Animal Care and Use Committee approved all experiments involving animals. Mice were on a 12-h light–dark cycle. Food (lab diet 5001) and water were available ad libitum. Housing rooms were maintained at 20–26 °C and relative humidity was 30–70%. Three transgenic mouse models were used in this work, each harboring the human C9ORF72 HRE and varying degrees of flanking sequence and copy numbers: BAC111, BAC112 (Dr. Robert Baloh, Novartis, Basel, formerly Cedars Sinai, Los Angeles), and C9-500 (Dr. Laura Ranum, U Florida Genetics Institute, Gainesville). BAC111 mice were generated in our laboratory and express approximately 600 repeat motifs within a truncated human C9ORF72 gene (from exons 1–6), and heterozygous mice express 6–8 copies and are described in greater detail in ref. 45. BAC112 mice express human C9ORF72 transgene with 550 repeat motifs, and heterozygous mice have 16–20 copies of the transgene[75]. The C9-500 C9ORF72 BAC-transgenic mouse model also expresses the whole human C9ORF72 gene, and heterozygous mice have one copy of the gene, with an expansion of 500 repeats[47]. For all experiments, heterozygous C9ORF72 BAC-transgenic mice were used. These were generated by crossing homozygous C9ORF72 mice with WT mice. To generate BAC111/Cas9 mice, homozygous BAC111 mice were crossed with homozygous B6J.129(Cg)-Gt(Rosa)26Sor^tm1.1(CAG-cas9*,-EGFP)Fezh/J (obtained from the Jackson Laboratory) resulting in heterozygosity of both BAC111 C9ORF72 and Cas9.

### Stereotaxic striatal brain injections

Mice were randomized into treatment groups, having equal numbers of males and females per group. Adult mice between ages of 8–16 weeks were anesthetized using Ketamine/Xylazine. Mice were then restrained on "Just for mouse stereotaxic instrument" (Stoelting). Two holes were drilled bilaterally at coordinates X = +/−2 mm, Y = +1 mm, Z = −2.5 mm, applying Bregma as a point of origin. Using a 10 µl Hamilton Syringe 10 µl, gastight model, 1701 RN and a removable 33-gauge Small Hub Needle (Hamilton Company), 5 µL were injected on each side at a rate of 0.5 µL/min. Mice were injected with $2 \times 10^{10}$ vector genomes (VG) of AAV9-Cas9 and $7 \times 10^9$ VP of the gRNA vectors or Rosa control per striatal side, which is $3.8 \times 10^{10}$ VG in total for both striata per mouse and $1.65 \times 10^{12}$ VG/kg. Animals were then administered three doses of pain medication Ketophen and Buprenorphine: directly after waking up from anesthesia and at 24- and 48-h post-surgery. Tissues were harvested eight weeks after injection for use in DNA, RNA, and protein analysis. Prior to harvest, mice were subjected to cardiac perfusion with phosphate-buffered solution (PBS). In all, 3 mm striatal punches were freshly taken using a disposable Biopsy punch with plunger (Integra) and then flash-frozen in liquid nitrogen and stored at −80 °C. For FISH, brains were frozen in the Optimal Cutting Temperature (OCT) compound and cryosectioned.

### Cell culture

For plasmid screening, HEK293T cells were seeded (1.2 E5 cells) in a 24-well plate, maintained with DMEM media (11995-073; Gibco), 10% Fetal Bovine Serum FBS (F-0926 Sigma-Aldrich) and 1% penicillin–streptomycin (30-001-CI; Corning). HEK293T cells were authenticated as they were purchased from ATCC (CRL-3216™). The cells were transfected after 24 h, using jetPRIME Reagent (Polyplus) and 1 µg of each plasmid DNA. Forty-eight hours post-transfection, cells were rinsed with PBS and collected for further analyses.

For primary neuron culture, homozygous C9ORF72 mice were crossed to wild-type mice to produce heterozygous progeny. Embryos were harvested at day 16 with cortices subsequently dissected in 4 °C Hank's Buffered Salt Solution (HBSS, Cellgro) under dissection microscope. The cortices were stored in HBSS until processing. Cells

were then dissociated for 15 min in 3.5 ml of 0.05% Trypsin (Invitrogen), with gentle inversion at 7.5-min intervals. After 15 min, Trypsin dissociation was inactivated by the addition of 7.5 ml of Dulbecco's Modified Eagle Medium (DMEM, Gibco-ThermoFisher) + 10% Fetal Bovine Serum (FBS) and homogenized via pipetting. This mixture was then added to a new 50 ml falcon tube through a 40-μm filter. An additional 10 ml of DMEM + 10% FBS was added through the filter to collect any remaining cells. Cells were spun down via centrifugation, and the pelleted cells were resuspended via pipetting in 20 ml of pre-warmed (37 °C) Neurobasal medium (Gibco-Thermofisher). After counting, Neurobasal was added to desired seeding concentration and supplemented with 20 μl/ml B27 (Invitrogen), 10 μl/ml Pen/Strep (Invitrogen), and 10 μl/ml Glutamax (Invitrogen). The cells were seeded on 6-well plates (Falcon/Olympus) at 1,000,000 cells/well in a volume of 2 ml per well. Cells were also seeded on 4-chamber CC2 glass slides (LAB-TEK) at 300,000 cells/chamber in a volume of 800 μl per chamber. Both the plates and chamber slides with slightly swirled immediately after seeding in order to prevent clumping. The plates were pre-coated with poly-d-Lysine (Invitrogen) at room temperature overnight, washed three times with ddH2O and dried before seeding. Cells were treated on DIV 4 post-plating by removing half of the media and replacing it with new media (pre-warmed and supplemented with B27, Pen/Strep, and Glutamax at previously described concentrations) containing the experimental vectors (AAV9-Cas9: MOI:30.000, AAV9-gRNA2,3 and AAV9-gRNA2,4 vector: MOI:20.000, with a total MOI:50.000). On day 10, the cells were harvested for additional experiments or fixed for FISH staining.

## iPSC cell lines and motor neuron differentiation

Patient-derived iPSCs were generated from ALS/FTD patient lymphocytes carrying an expansion of 450 repeats in C9ORF72, which was obtained from blood of consenting patient within our patient population under strict IRB-approved protocols at the University of Miami Miller School of Medicine. iPSCs were differentiated into motor neurons as follows: iPSCs were maintained in Matrigel-coated flasks and mTeSR® medium (StemCell Technologies). For neural induction, iPSCs were dissociated and transferred to low attachment (poly-HEMA) flasks and cultured in suspension with neuroinduction medium for 21 days. Neurospheres were then seeded onto poly-L-ornithine/Laminin (PLO)-coated plates to promote attachment and maintained in neuronal maturation medium for an additional 21 days[92].

## AAV transduction of C9ORF72 iPSCs

Matrigel-coated 12-well plates were seeded with undifferentiated iPSCs ($3.5 \times 10^5$ cells/well). The following day, cells were co-transduced in a volume of 250 μL/well with AAV9–SpCas9 and AAV9-gRNA2,4 vectors at a multiplicity of $3 \times 10^5$ and $1 \times 10^5$ AAV genome copies per cell, respectively. After 6 h, an additional 1 mL of medium was added to each well without removing viral vectors and maintained for 3 days at 37 °C and 5% $CO_2$ in a humidified incubator. Transduced iPSCs were collected using gently dissociated reagent and seeded at low density in Matrigel-coated 6-well plates. After 3 days, single colonies were manually selected, expanded, and analyzed for genome editing. To ensure the clonality of iPSC lines, cultures were subcloned by titration and manual selection.

## AAV transduction of differentiated C9ORF72 iPSC motor neurons

After 42 days of neurodifferentiation, iPSC motor neurons were transferred to 12-well PLO-coated plates ($3.5 \times 10^5$ cells/well) and allowed to recover for 7 days. Neuronal cultures were co-transduced in a volume of 250 μL/well with AAV9–SpCas9 and AAV9-gRNA2,3 or AAV9-gRNA2,4 vectors at a multiplicity of $3 \times 10^5$ and $1 \times 10^5$ AAV genome copies per cell, respectively. After 6 h, an additional 1 mL of

medium was added to each well without removing viral vectors and maintained for 7 days at 37 °C and 5% $CO_2$ in a humidified incubator.

## Generation and AAV transduction of brain organoids

Cultures of C9ORF72 iPSCs were dissociated, transferred to low attachment flasks, and maintained in brain organoid induction medium (1× DMEM F12, 1× N2 supplement, 10 mg/ml heparin, 1× penicillin/streptomycin, 1× MEM non-essential amino acids, 1× glutamax, 1 μM CHIR99021, 1 μM SB-431542, and 100 μg/ml Primocin)[62]. After 5 or 17 days, embryoid bodies were transduced with 15 μL of AAV9-Cas9 ($5.5 \times 10^{12}$ genome copies/mL) and 5 μL AAV9-gRNA2,4 ($8.5 \times 10^{12}$ genomes/mL) in a volume of 500 μL for 30 minutes. An additional 3 mLs of medium used to suspend and transfer embryoid bodies to a 12-well plate. On day 32, organoids were harvested by centrifugation and genomic DNA was isolated for analysis of genome editing.

## Droplet Digital PCR (ddPCR)

Droplet Digital PCR (ddPCR) analysis was conducted on samples from in vitro (primary neurons) and in vivo (mouse striatal injection) experimentation. Frozen tissue samples were homogenized using a TissueLyser II (Qiagen) before a total RNA extraction with TriZol (Life Technologies, USA) in accordance with the TriZol reagent user guide protocol (Invitrogen). For in vitro samples, 1 ml of Trizol was directly added to frozen harvested primary neurons, homogenized via pipetting, with RNA subsequently extracted following the previously described TriZol reagent protocol. Reverse transcription to cDNA was performed using a High capacity RNA-to-cDNA Kit (Applied Biosystems-Thermofisher) in accordance with their quick reference guide protocol. ddPCR was then performed on the cDNA to assess the absolute concentration of C9ORF72 transcripts. Master mixes for ddPCR were generated containing 11 μl/sample of ddPCR Supermix (no dUTP) (Bio-Rad, USA), 1.1 μl/sample of C9ORF72 transcript-specific FAM labeled probe, 1.1 μl/sample of TATA-box Binding Protein (TBP) specific HEX labeled probe as a housekeeping probe, and 8.8 μl of cDNA diluted in ddH20. Sample droplets were generated by loading 20 μl of sample and 70 μl of ddPCR oil in to a DG8 cartridge (Bio-Rad, USA) and placing the cartridge in a QX100 droplet generator (Bio-Rad, USA). 40 μl of the generated droplets were added to each well of a ddPCR plate and placed in a PCR thermocycler under the following conditions: 95 °C for 10 min, 40 cycles of {*95 °C for 30 s, *55 °C for 1 min}, *98 °C for 10 min, hold at 4 °C. (*) denotes a 30% RAMP. Following PCR, the ddPCR plate was read in a QX100 Droplet Digital PCR reader (Bio-Rad, USA) and analyzed using QuantaSoft software.

## Western blot

Radioimmunoprecipitation Assay (RIPA) buffer and cOmplete mini-proteinase inhibitors (1 tablet per 10 ml extraction solution, Roche) was used to homogenize striatal tissue and cells. Equal amounts of protein lysates, as measured by BCA Assay (Thermo Scientific, USA), were run on Novex 12% Tris-Glycine gels (Life Technologies, USA) using Tris-Glycine SDS running buffer (Invitrogen, USA). Protein was transferred to nitrocellulose membranes using an i-Blot transfer device (Invitrogen, USA). Membranes were blocked for 1 h at room temperature with Odyssey Blocking Buffer (LiCor, USA) before being probed overnight with primary antibodies against C9ORF72 (rabbit, polyclonal, 1:1000, 22637-1-AP, Thermo Scientific, USA). Infrared labeled secondary antibodies were applied (WesternSure goat anti-rabbit HRP secondary antibody, 1:5000, 926-80011, LI-COR Biotechnology, USA), and blots visualized. The band intensity was quantified using the Odyssey Infrared imaging system (LiCor, USA).

## Poly-GP and Poly-GR immunoassay

Both primary neuron and mouse tissues were lysed in Radio-immunoprecipitation Assay (RIPA) buffer and a cOmplete protease inhibitor (Roche). Tissues were lysed using a hand lyser. Then, samples

were centrifuged at 16,000 × g for 20 min and supernatant collected. Total protein concentration of lysates was determined by a BCA assay (Thermo Scientific, USA). Poly-GP and Poly-GR levels in lysates were measured using a sandwich immunoassay that utilizes Meso Scale Discovery (MSD) electrochemiluminescence detection technology[37,48]. Lysates were diluted in tris-buffered saline (TBS) and tested using equal concentrations for each sample in duplicate wells. Response values corresponding to the intensity of emitted light upon electrochemical stimulation of the assay plate using the MSD QUICKPLEX SQ120 were acquired and background-corrected, using the average response from non-transgenic mice.

### RNA fluorescence in situ hybridization (FISH)

Primary cortical neurons grown on 4-chamber CC2 glass slides (LAB-TEK) or cryosections (20 µm) were fixed in 4% PFA for 20 min. Samples were then washed with DEPC-treated PBS containing 0.1% Tween-20 for 20 min five times and prehybridized for 1 h at 55 °C in 2× DEPC-treated saline sodium citrate (SSC) containing 40% formamide, 0.1% Tween-20, 50% of dextran sulfate, 50 µg/ml heparin and 1 mg/ml heat-denatured salmon sperm. Samples were hybridized overnight at 55 °C in a pre-hybridization buffer containing a 1:500 dilution of 1 µM sense GGCCCC$_4$ DNA probe with Cy3 5′ end tag. After hybridization, the samples were washed in a pre-warmed mix 1:1 of hybridization buffer and 2× SSC for 30 min at 55 °C. Followed by two washes in pre-warmed 2× SSC for 30 min at 55 °C, 2 washes with 0.2× SSC for 30 min at 55 °C, and two washes in 1× PBS + 0.1% Tween-20 for 30 min, at room temperature. Nuclei were stained by 4′,6-diamidino-2-phenylindole (DAPI, 1:10,000) for 5 min, and were then incubated in 0.5% Sudan Black in 70% ethanol for 7 min to reduce autofluorescence. Lab Vision Perma-Fluor (Thermo Scientific Shandon, USA) was used to mount glass coverslips. For quantification, z-stacks of the cultured primary neurons and external pyramidal layer of brain tissues were acquired by a Leica DM5500 B microscope with a Leica DFC365 FX camera. A total magnification of ×100.8 (×63 oil lens *1.6× camera) was used to generate the images, which were maximally projected, and foci were counted manually in +300 cells per slide.

### RNAscope

RNAscope in situ hybridization (ish) Fluorescent Multiplex Assay was performed as instructed by advanced cell diagnostics (ACD). Brains were embedded in OCT, frozen and sectioned at 20 µm. Sections were fixed in 4% PFA for 1 hour and dehydrated in 50% ethanol (5 min), 70% ethanol (5 min) and 100% ethanol (10 min) at room temperature. The slides were air dried and protease IV reagent was added to each section for 30 min. Slides were washed in PBS at room temperature and treated with a mixture of Channel 1 (GFP, 409011, ACD, USA) and Channel 3 probes (SpCas9, 533931-C3, ACD, USA) at a ratio of 50:1 dilution for two hours at 40 °C. Following probe incubation, the slides were washed two times in 1× RNAscope wash buffer and submerged in AMP-1 reagent for 30 min at 40 °C. Washes and amplification were repeated using AMP-2, AMP-3, and AMP-4 ALT A (channel 1 = 488 nm, channel 3 = 647 nm) reagents with a 15-min, 30-min, and 15-min incubation period, respectively. Sections were counterstained with DAPI for 30 s at room temperature and cover-slipped using Lab Vision PermaFluor (Thermo Scientific Shandon, USA). Pictures of the staining were acquired by a Leica DM5500 B microscope with a Leica DFC365 FX camera. Total magnification of ×100.8 (×63 oil lens * 1.6× camera) was used to generate the images.

### PacBio No-Amp

Genomic DNA was extracted using the Monarch® HMW DNA Extraction Kit (New England Biolabs, T3050S and T3060S) using the manufacturer's protocol for iPSC samples, and a modified manufacturer's protocol for striatal tissues. Shortly, DNA was extracted from striatal tissues using the "low input" protocol. Tissues were

### Table 1 | PacBio No-Amp number of reads

| | iPSC | | C9-500 striatum | | |
|---|---|---|---|---|---|
| | C9 control | gRNA2,4 11-11 | PBS | gRNA2,3 | gRNA2,4 |
| #1 | 180 | 140 | 59 | 67 | 59 |
| #2 | | | 90 | 56 | 80 |
| #3 | | | 80 | 54 | 59 |

homogenized by pestle in 300 µl lysis buffer and 10 µl Proteinase K and then incubated at 56 °C for 10 min in a thermal mixer with 1400 RPM agitation and 35 min without agitation. Afterward, an additional 5 µl Proteinase K was added to the samples and incubated for 45 min without agitation. In total, 5 µl RNase A was added and mixed by inverting ten times and incubated for 10 min at 56 °C at 1400 RPM. Then, samples were centrifuged at maximum speed for 3 min, and the supernatant was transferred to a new tube and incubated on ice for 5 min. Next, 150 µl of Protein Separation Solution was added and mixed by inverting for 1 min. This was followed by maximum centrifugation at maximum speed for 20 min. The remainder of the DNA extraction was executed following the manufacturer's protocol. After DNA extraction, striatal DNA was reprecipitated using 0.1 volume 3 M sodium acetate and 2.5 volume ice-cold 100% ethanol at −20 °C overnight. The next day, samples were centrifuged at maximum speed at 4 °C for 30 min, and pellets were washed twice with 0.5 ml ice-cold 75% Ethanol, spinning at 4 °C for 10 min each time and resuspended in EB.

PacBio No-AMP was conducted following PacBio's "No-Amp Targeted Sequencing Utilizing the CRISPR-Cas9 System" manual. After inspecting the quality of the genomic DNA using a Femto Pulse system and Nanodrop, the SMRTbell library was prepared as follows: To reduce the amount of off-target molecules in the final SMRTbell library, the DNA (3 µg per sample) was dephosphorylated to prevent fragment ends from participating in the ligation reaction following the CRISPR–Cas9 Digestion step. Then, the dephosphorylated DNA was digested with Cas9 and gRNAs adjacent to the region of interest (as published in the manual, C9orf72.DC.1: TTGGTATTTAGAAAGGTGGT; and C9orf72.DC.2: GGAAGAAAGAATTGCAATTA), after which the DNA was purified using AMPure PB beads. Next, a capture adapter was ligated to the blunt ends and captured using AMPure PB beads. Then, failed ligation products and gDNA fragments were removed with a nuclease treatment. Finally, an additional treatment of the SMRTbell library with trypsin was used to remove enzymes during the AMPure PB bead-purification step.

For sequence preparation, a standard PacBio sequencing primer was annealed to the enriched SMRTbell templates, and a Sequel II DNA polymerase 2.2 was bound to the primer-annealed SMRTbell templates. This complex was purified with AMPure PB beads and loaded onto a SMRT Cell, and sequenced on a Sequel II System using an immobilization time of 4 h and movie time of 30 h.

HiFi long reads were produced using the standard demultiplexing and circular consensus tools in SMRT Link v10.2, with the exception that the "--all" input option was specified during circular consensus such that a read was output for every ZMW. For each sample, the reads were then aligned to human reference GRCh38 using the pbmm2 aligner, and off-target reads were removed. Reads spanning the targeted repeat region were assessed for the presence or absence of the inserted repeat sequence in one of two ways based on whether or not they spanned the entirety of the repeat region reference coordinates (chr9: 27,573,485–27,573,546). For reads spanning the region (Table 1), repeat lengths were determined based on the number of inserted bases in the region (via parsing CIGAR strings and extracting the corresponding read sequence). For reads only anchored on one side, and thus are soft-clipped in the repeat region, a manual inspection of the clipped sequence was

## Table 2 | Primers used for UDiTaS

| | |
|---|---|
| 3742_UDiTaS_i5_ common | AATGATACGGCGACCACCGAGATCTACAC |
| 3895_Uditas_g2Nested_C9ORF72 | TCTCTCCCCACTACTTGCTCTCACAGTACTC |
| 3743_C9ORF72_gRNA2_i7 | GTGACTGGAGTTCAGACGTGTGCTCTTCCGATCTCGCTGAGGGTGAACAAGAAAAGACCTG |
| UDiTaS adapter top i501 | AATGATACGGCGACCACCGAGATCTACAC**TATAGCCT***NNWNNWNNN*TCGTCGGCAGCGTCAGATGTGTATAAGAGACAG |
| UDiTaS adapter top i502 | AATGATACGGCGACCACCGAGATCTACAC**ATAGAGGC***NNWNNWNNN*TCGTCGGCAGCGTCAGATGTGTATAAGAGACAG |
| UDiTaS adapter top i504 | AATGATACGGCGACCACCGAGATCTACAC**GGCTCTGA***NNWNNWNNN*TCGTCGGCAGCGTCAGATGTGTATAAGAGACAG |

The bold sequence represents 5p index sequence and the italic sequence represents the UMI.

## Table 3 | UDiTaS number of reads

| | HEK293T | | | | iPSC | | C9-500 striatum | | | |
|---|---|---|---|---|---|---|---|---|---|---|
| | **PBS** | **ROSA** | **gRNA2,3** | **gRNA2,4** | **C9 control** | **gRNA2,4 11-11** | **PBS** | **ROSA** | **gRNA2,3** | **gRNA2,4** |
| #1 | 2220 | 318 | 8149 | 2999 | 34,192 | 76,114 | 18,327 | 16,670 | 47,943 | 70,315 |
| #2 | 863 | 1142 | 4819 | 5697 | 45,784 | 78,449 | 29,500 | 19,244 | 41,089 | 67,761 |
| #3 | 405 | 616 | 4575 | 3335 | 41,522 | 16,100 | 31,847 | 25,003 | 56,987 | 67,457 |
| #4 | 448 | 628 | 8201 | 3651 | | | 30,028 | 20,677 | 31,497 | 48,168 |

performed to identify whether or not it contained the inserted repeat sequence. Editing efficiencies were then determined from the fraction of reads without the inserted repeat sequence.

### UDiTaS library preparation and analysis

Transposome was assembled according to published protocol[43]. Briefly, oligonucleotides containing unique molecular identifiers (UMIs) and i5 barcodes obtained from IDT were annealed and loaded into purified TN5. Library preparation was performed using a nested PCR approach[83]. Briefly, 1 μL of transposome was used to tagment 200 ng of genomic DNA at 55 °C for 7 min. After tagmentation, 1 μL of 0.2% SDS was added to quench any remaining enzyme. The tagmented genomic DNA was used for the first round of PCR for library preparation with a common primer (located within the transposon) and a C9ORF72-specific nested primer for ten cycles (Table 2). 0.9× AMPure beads were used to purify the products. All of the products from the initial PCR were used in the second round of PCR with common primer and internal C9ORF72 primer for 15 cycles. In all, 0.9× AMPure beads were used to clean up the products for the third round of PCR which used UDiTaS_i5_-common and i7 barcode primers for 20 cycles. A final purification was performed using 0.5×/0.35× for double size selection to yield a size distribution of ~200–1000 bp in the final library. The libraries were sequenced on an Illumina MiniSeq for 2 ×150 cycles.

The analysis was performed using the UDiTaS v1.0 workflow described in ref. 43 with modifications to allow compatibility with the MiniSeq platform[83]. Paired-end libraries were demultiplexed with bcl2fastq (Illumina), masking UMIs (12 nts 1–9). The undemultiplexed UMIs were stored in a single FASTQ masking R1, I1, I2 nts 10–17. The demultiplexed R1 and R2 FASTQs were paired via UMIs with fastq-pair hash lookup tool[93]. Gzipped FASTQs were added to individual UDiTaS compatible sample directories. Reference FASTA and 2-bit files were obtained from UCSC table browser[94]. The analysis[43] was performed inside umasstr/UDiTaS docker container, skipping the demultiplexing step (Table 3).

### Statistics and reproducibility

RNAscope and FISH experiments were blinded during execution and analysis. Animals were randomized into treatment groups, having equal numbers of males and females per group. Unless otherwise indicated, data were analyzed by two-tailed Student's *t* test, or one-way or two-way analysis of variance (ANOVA) followed by post hoc analyses for multiple comparisons with an appropriate test using GraphPad Prism version 9.0.0 software. Statistical significance was assumed with a *P* value <0.05 (*), *P* < 0.01 (**), *P* < 0.001 (***) and *P* < 0.0001 (****). Bars in graphs represent mean ± SEM.

### Reporting summary

Further information on research design is available in the Nature Research Reporting Summary linked to this article.

## Data availability

The sequencing datasets generated and analyzed in this study are available in the NIH Sequence Read Archive via BioProject accessions PRJNA871245 and PRJNA870226. Source data are provided with this paper.

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

## Acknowledgements

We thank the Mutagenesis Core Facility at UMASS Medical School for the design of gRNAs and in silico off-target effects analyses. We further thank G. Gao, J. Xie, and Q. Su at UMass Medical School who supervised the vector production and D. Wang who provided us with the SpCas9 plasmid. Furthermore, we are grateful to Eric Wieben and the Genome

Analysis Core at the Mayo Clinic for performing the PacBio No-AMP experiments. Special thanks to Melany Hirsch for her assistance. R.H.B. and C.M. received support from NINDS NS088689. In addition, R.H.B. was funded by R01 NS111990 and R01 NS104022, the ALS Association, the Angel Fund for ALS Research, the Pierre L. de Bourgknecht ALS Research Foundation, ALS Finding A Cure, ALSOne, the Cellucci Fund, and the Max Rosenfeld ALS Research Fund. R37NS057553 and R01NS101986 and the Target ALS Foundation grant were rewarded to F.B.G.

## Author contributions

K.E.M. led the design, analysis, interpretation of the study, and prepared the manuscript. C.M. conceived of the project and supervised all aspects of its execution and analysis. gRNAs were designed by M.H.B. Testing of individual guide RNAs in HEK293T cells was executed by H.N. Cloning and prepping of constructs were done by K.E.M, H.N., A.A., and M.B. PCRs were performed by K.E.M., H.N., and A.A. Primary neuron experiments, mouse surgeries and FISH staining were executed by K.E.M., A.A., and N.P.F. Experiments involving transcript levels were performed by K.E.M., N.P.F., and M.T.B. Western blots were done by K.E.M. Poly-dipeptide levels were detected by K.E.M., G.K., and T.F.G. Immunohistochemical stainings were executed by K.E.M., A.K., and N.H. iPSC, iMN and brain organoid experiments were executed by Z.Z., R.E., N.S.A., M.J.R., and M.R. InDel and off-target analyses were performed by K.E.M., A.E., and T.R. PacBio No-Amp analyses were executed by Z.D.S and K.E.M., oversight by J.P.A.K. UDiTaS analyses were performed by T.R., C.K., and K.E.M. Lastly, R.H.B., Z.Z., E.J.S., F.B.G., M.H.B., L.P., and C.M. have aided with scientific input, and N.P.F, Z.Z., R.H.B., F.B.G. M.B., and E.J.S. have aided with editing.

## Competing interests

C.M., A.A., and UMASS Medical School are inventors on the patent (number US 2020/035471, still pending) for the gRNA sequences in this work and may be entitled to royalty payments in the future. The remaining authors declare no competing interests.

## Additional information

**Katharina E. Meijboom** [1,2], **Abbas Abdallah**[1], **Nicholas P. Fordham** [1], **Hiroko Nagase**[1], **Tomás Rodriguez** [3], **Carolyn Kraus** [3], **Tania F. Gendron** [4], **Gopinath Krishnan**[2], **Rustam Esanov**[5], **Nadja S. Andrade**[5], **Matthew J. Rybin** [5], **Melina Ramic**[5], **Zachary D. Stephens**[6], **Alireza Edraki**[3], **Meghan T. Blackwood**[1], **Aydan Kahriman**[2], **Nils Henninger**[2], **Jean-Pierre A. Kocher** [6], **Michael Benatar**[7], **Michael H. Brodsky**[8], **Leonard Petrucelli**[4], **Fen-Biao Gao** [2], **Erik J. Sontheimer** [3], **Robert H. Brown** [2], **Zane Zeier** [5] ✉ & **Christian Mueller** [1] ✉

[1]Horae Gene Therapy Center, University of Massachusetts Medical School, Worcester, MA 01605, USA. [2]Department of Neurology, University of Massachusetts Medical School, Worcester, MA 01605, USA. [3]RNA Therapeutics Institute and Program in Molecular Medicine, University of Massachusetts Medical School, Worcester, MA 01605, USA. [4]Department of Neuroscience, Mayo Clinic, Jacksonville, FL 32224, USA. [5]Department of Psychiatry & Behavioral Sciences, Center for Therapeutic Innovation, University of Miami Miller School of Medicine, Miami, FL 33136, USA. [6]Department of Quantitative Health Sciences. Mayo Clinic, Rochester, MN 55905, USA. [7]Department of Neurology, University of Miami Miller School of Medicine, Miami, FL 33136, USA. [8]Department of Molecular, Cell and Cancer Biology, University of Massachusetts Medical School, Worcester, MA 01605, USA. ✉e-mail: zzeier@med.miami.edu; Christian.mueller4@sanofi.com

