## [Peer Review File · Nature Communications]

CRISPR/Cas9-Mediated Excision of ALS/FTD-Causing Hexanucleotide Repeat Expansion in C9ORF72 rescues major disease mechanisms in vivo and in vitroREVIEWER COMMENTS

Reviewer #1 (Remarks to the Author):

This is a review of the manuscript by Meijboom et al entitled "CRISPR/Cas9-Mediated Excision of ALS/FTD-Causing Hexanucleotide Repeat Expansion in C9ORF72 rescues major disease mechanisms in vivo and in vitro". Hexanucleotide repeat expansions in the C9ORF72 gene are the most frequent cause of familial ALS and FTD, and there are no current treatments for affected patients. Here the authors employ an innovative AAV/CRISPR-Cas9 approach to deliver spCas9 and two sgRNAs on either side of the repeat, thereby inducing cuts that flank the repeat and its secondary deletion. They test this system across a range of cell types including patient-derived C9ORF72 mutant iPSCs and derived neurons, and an in vivo mouse C9ORF72 BAC mouse model, showing that AAV delivery causes a reduction in disease-relevant pathology (RNA foci, dipeptide repeats (DPRs), and in some cell types restoration of normal levels of expression of the C9 protein.

In general, this is a very interesting and important manuscript, providing critical proof of concept data regarding the potential of CRISPR/Cas9 + AAV therapeutic modalities for patients with C9ORF72 mutations. Below I raise several mainly technical concerns that could nevertheless influence interpretation of the data and/or suggest alternative strategies to further improve editing. Once addressed, I would fully support publication of this manuscript.

Major concerns

Co-transduction efficiencies: As the authors note, due to packaging size restrictions of AAV, it was necessary to deliver Cas9 and 2x sgRNAs through co-transduction with different vectors. What percentage of individual neurons in vivo and in vitro actually co-express the Cas9 and sgRNAs? My understanding is that the frequency of co-transduction of two different AAVs in individual cells is much lower than transduction of a single AAV. If this is the case, it will be difficult to determine the efficiency of editing versus the efficiency of dual-AAV delivery. If co-transduction is indeed low, it could also prompt simple improvements to the strategy of AAV transduction (e.g. transduce in two separate rounds, with each round containing a different AAV, vs a single injection with both AAVs).

- o I recommend that the investigators co-stain their mouse brain sections, iPSC neurons, and organoids with anti-Cas9 (or anti-HA) and determine the fraction of cells that are GFP + (i.e. express sgRNAs) and Cas9 +.

- o It would be additionally important to look at this fraction of double-positive neurons and quantify what fraction of such neurons lose pathologic hallmarks of repeat overexpression (DPRs, RNA foci).

Unclear editing efficiencies and frequency of on-target genomic rearrangements

- o The authors use PCR-based assays to assess editing efficiencies in a number of experiments (e.g. Fig 2b, fig 3b). These assays are non-linear and only semi-quantitative at best. In addition, it is now recognized that on-target genome rearrangements occur at unexpectedly high frequencies in iPSCs during Cas9-mediated editing, and if not directly assayed are often missed (PMID 30837594).

- ♣ Improved quantification and confirmation of editing through southern blotting in at least one of their cellular models (ideally patient-derived human iPSC neurons or organoids) would be superior.

- ♣ As a complementary technique, NGS sequencing of amplicons generated through primers flanking the edited site would enable them to identify allele-specific SNPs. Identification of allele-specific SNPs would allow the authors to distinguish between desired editing events and large "on-target" rearrangements in clones that could mimic the desired edit via gel electrophoresis but actually be due to deletion of primer binding sites on one of the alleles. An example of such an odd rearrangement may be clone 17,

Fig 4b, but other clones that appear to be successfully edited based on gel electrophoresis could theoretically also harbor on-target rearrangements.

Incomplete evaluation of iPSC/iPSC neuron editing events and non-clonality assayed iPSC "clones"

o Testing their AAV system in patient-derived iPSCs with a single copy of bona-fide C9ORF72 repeats at the endogenous locus are key experiments for this manuscript. However, in Figure 4, the authors only demonstrate excision of the repeat in a single subclone (11-11), and in fact demonstrate that a "clone" in Fig4b was clearly polyclonal. In my opinion, this is likely because their protocol for isolating iPSC "clones" is highly biased toward selecting polyclonal clusters of cells. In their methods, they appear to pick clones post-dissociation after 72h. The doubling time of most iPSC lines is ~12-24h, so it is virtually impossible to pick a true clone of isogenic iPSCs 3d post-dissociation.

o I would recommend that they repeat experiments performed in Fig 4b-d, modifying their protocol to improve frequency of isolating true single clones. The latter could be achieved through improved dissociation, FACS-based sorting of single cells into wells, or serial dilutions of dilute cell populations – coupled with waiting for a longer period of time prior to picking single clones – to improve their chances of identifying isogenic clonal populations. This would also allow them to assay additional clones (which I think is necessary) to demonstrate changes in poly GP levels and repeat prime PCR assays.

Minor concerns

- The authors should note in the discussion that AAVs will not target certain cell types that may nevertheless be relevant to C9 pathogenesis, for example microglia.**
- The editing frequency in organoids is modest at best; this should be pointed out. Appears likely due to relatively poor AAV transduction efficiencies based on GFP, likely compounded by issues that I raised regarding dual-transductions in single cell (if frequencies for the single GFP AAV are low, the chances of dual transduction are likely even lower).**
- The authors should consider mentioning that new versions of Cas9 and Cas9-like nucleases are much smaller than traditional spCas9, and that use of these nucleases could enable single AAV vector delivery of nucleases and sgRNAs. Eg. PMID: 32675376**
- Fig 4h y-axis text is cut-off**

Reviewer #2 (Remarks to the Author):

Meijboom et al developed AAVs to excise the pathogenic repeat expansion in C9orf72 models using CRISPR. The validated their approach in primary neurons, iPSC models and most importantly, also directly in two different BAC transgenic mouse models. This could be a very interesting therapeutic option in the future. The manuscript is well written and the data seems solid. For publication in Nat Comm some issues need to be addressed. Quantifying the efficacy of gene editing especially in mice seems most relevant.

Major points:

- Fig 2a: Homozygous x WT does not fit to the experiments with Cas9 transgenic mice**
- Line 151: The authors used an MOI of 50000 in Cas9 expressing neurons. This high MOI will certainly no be achievable in patients. To get more therapy-relevant information the MOI should be titrated and editing efficacy quantified.**
- Fig 2c: non-transgenic controls should be included to determine the background level and thus the extent of rescue. It would also be interesting to the community to compare the poly-GP level ins BAC111 and C9-500 neurons**
- Fig 2d: Since foci are barely visible in the PBS BAC111 mice, non-transgenic controls should be shown. Are the foci only enlarged with gRNA2,3 or also gRNA2,4? This finding is very hard to explain. Is it associated with more intronic RNA on qPCR level? Were enlarged foci also observed in the treated BAC111 mice?**
- Fig 2g: The original western blots should be shown, because C9orf72 antibodies are**

notoriously bad.

-Fig 2b/3b: Can the edited amplicon be quantified using real-time PCR to see the relative amount of editing under different conditions? Using unedited and edited iPSC lines from Fig 4 as a standard it might be possible to even quantify editing efficacy in absolute numbers.

-Fig 3c/d: Background in non-transgenic mice should be shown for GP and GR assays. The lack of correlation of GP and GR could be explained by poor performance of the GR assay. Have the GP and GR measurements been done on different animals to attribute it to injection issues? GP assays should be included for BAC111 mice. Absolute levels of GP/GR should be compared between the two mouse lines. Immunostaining should be shown to support immunoassay data.

-Fig 3: Immunostaining for Cas9 should be shown to estimate transduction and spread in the CNS. The authors should attempt to quantify the percent of edited cells in vivo to support therapeutic utility. This might be done using qPCR or maybe DNA FISH of the edited (or unedited) locus.

-In all qPCR experiment the authors infer changes in transcript V2 arguing it cannot be quantified because primers would interfere with the gRNA target sites. However, V2 can be easily quantified directly using primers targeting the 3' end of the mRNA (e.g. PMID: 24559645 or commercial probes). This should be done.

Minor points:

-Line 252: poly-GP is the most abundant poly-dipeptide from the antisense strand, but not in general.

-Line 274/275: at least V3 did change in the primary neurons (see Fig 2f)

-Line 325: figure 4g?

Reviewer #3 (Remarks to the Author):

In the manuscript by Meijboom KE et al., the authors used CRISPR-Cas9 system to excise expanded hexanucleotide repeats in C9ORF72 gene responsible for ALS/FTD. They demonstrated successful editing of this locus in various cellular models (human and mouse), brain organoids as well as in three mouse models. As a result the RNA foci and toxic polypeptides were reduced, and the transcript and protein levels were restored. The authors suggest that this strategy can be used as a therapeutic approach for ALS and FTD.

What are the noteworthy results? The efficient excision of the mutant repeat tract with the use of CRISPR-Cas9 system has been already demonstrated for different STR-containing loci, including DMPK with a CTG expansion, HTT with a CAG expansion and even C9ORF72; therefore the approach presented in the manuscript is not original. The authors used the most basic CRISPR-Cas9 system tools, such as *Streptococcus pyogenes* Cas9 and a pair of unmodified gRNAs. Currently a genome editing field offers more advanced tools, more specific, safer, or smaller which would be better for therapeutic approaches. Although promising, the therapeutic application of the strategy presented in the manuscript would require more research including more detailed analysis of off-targets, improving delivery issues, etc. Conventional SpCRISPR-Cas9 can be nonspecific and further improvements of this strategy are necessary. In addition the author's findings that excision of expanded repeats rescues major disease mechanisms in vitro is not completely novel. For example, Selvaraj et al., (doi: 10.1038/s41467-017-02729-0) demonstrated that excision of the G4C2 repeat expansion by CRISPR-Cas9 system reverses RNA foci pathology in mutant motor neurons.

Nonetheless, it is an interesting study which extends other reports and demonstrates that the CRISPR-Cas9 technology can be successfully used to decrease pathological hallmarks of ALS/FTD. The manuscript is well written, the data are generally strong and well presented. The number of models is large. However, some conclusions are not supported by the data and some aspects needs clarification:

Major:

1. Additional evidence is needed to support a conclusion that the presented strategy

does not generate off-target effects. In the paper "Unexpected Mutations by CRISPR-Cas9 CTG Repeat Excision in Myotonic Dystrophy and Use of CRISPR Interference as an Alternative Approach" (DOI: 10.1016/j.omtm.2020.05.024) the authors used similar strategy to excise long CTG repeats in DM1. The authors used a conventional Cas9 nuclease and Cas9 nickase and demonstrated that the formation of the RNA foci was markedly reduced in patient-derived fibroblasts. In addition they observed a considerable amount of unintended off-target mutations and genomic rearrangements using high-throughput genome-wide translocation sequencing. Therefore, more detailed analysis of unintended events (especially genomic rearrangements) should be performed in this study using unbiased method to support the idea of using this approach for therapeutic purposes.

2. The authors assumed that excision of ex2 present in V2 mRNA (the most highly expressed C9ORF72 variant) has no influence on the overall mRNA and protein levels. Perhaps it is so, but a more "elegant" approach would be the excision of repeats without disruption of the ORF. Was it impossible? (splicing disruption, lack of a PAM sequence within 35bp flanking sequence,...)?

3. The authors used biased PCR-based methods to analyze the quality and quantity of edits. This excludes the possibility of detection of larger deletions, expansions.

3. Efficient excision was demonstrated using many different models. Did the authors expect different DNA repair results in these models? (different quality or quantity of edits?).

4. Fig.2d: how to explain that the number of foci is higher in BAC111 model treated with gRNA 2,3? The conclusion presented by the authors (lines 199-201) is not convincing, since the control cells (PBS) do not have foci (at least in the picture shown).

5. Fig.3d: as above, I am not convinced by the explanation of poly-GR levels in C9-500 mice treated with gRNA 2,3 (lines 263-266). The authors did not exclude the possibility of genomic rearrangements, or other unintended events induced by excision of a large DNA sequence. In addition, since RAN proteins production can be higher in stress conditions and AAV9-CRISPR-Cas9 system injection induces stress, PBS control is probably not the best choice for this experiment.

Minor points:

1. Dividing the text into sub-chapters would be helpful especially when so many different models are used.

2. Fig. 1b (lines 117-119), it is unclear if control was Untr (?) or plasmid, please explain

3. Fig. 1 description: NoE-F1 primer is not present on the Figure and its function is unclear

4. Line 92: please expand the abbreviation iMN,

5. Line 121: "efficient" instead of "efficiently"

6. Line 126, 136: "sanger sequencing", "sanger-sequencing"

7. Lines 129-130: please correct the sentence

8. Line 325: Fig.4g instead of 4e

9. Line 544: What does it mean that InDels up to 20bp were permitted? Please explain.

10. Line 551: please explain why AMACR locus was analyzed using different parameters; please add more details concerning off-target analysis and deep sequencing (the length of amplicons, depth, RPM,...)

Reviewer #1

Hexanucleotide repeat expansions in the C9ORF72 gene are the most frequent cause of familial ALS and FTD, and there are no current treatments for affected patients. Here the authors employ an innovative AAV/CRISPR-Cas9 approach to deliver spCas9 and two sgRNAs on either side of the repeat, thereby inducing cuts that flank the repeat and its secondary deletion.In general, this is a very interesting and important manuscript, providing critical proof of concept data regarding the potential of CRISPR/Cas9 + AAV therapeutic modalities for patients with C9ORF72 mutations. Below I raise several mainly technical concerns that could nevertheless influence interpretation of the data and/or suggest alternative strategies to further improve editing. Once addressed, I would fully support publication of this manuscript.

We appreciate this reviewer's statement that our work is innovative and important and offer the following responses to his comments.

Major points:

1-1. *Co-transduction efficiencies: What percentage of individual neurons in vivo and in vitro actually co-express the Cas9 and sgRNAs? My understanding is that the frequency of co-transduction of two different AAVs in individual cells is much lower than transduction of a single AAV. If this is the case, it will be difficult to determine the efficiency of editing versus the efficiency of dual-AAV delivery. If co-transduction is indeed low, it could also prompt simple improvements to the strategy of AAV transduction (e.g. transduce in two separate rounds, with each round containing a different AAV, vs a single injection with both AAVs).*

I recommend that the investigators co-stain their mouse brain sections, iPSC neurons, and organoids with anti-Cas9 (or anti-HA) and determine the fraction of cells that are GFP + (i.e. express sgRNAs) and Cas9 +.

We used RNAScope to stain for Cas9 and GFP (expressed by the gRNA vectors). We found that most cells in the striatum were transduced with both AAVs: 58% for gRNA2,3 and 69% for gRNA2,4 treated C9-500 mice (Figure 4a,b). This is in line with the decrease we found in toxic gain of function, polydipeptides (Figure 3c,d) and RNA foci (Figure 3e,f).

We agree that our dual vector approach, while expedient for testing multiple gRNAs, is not ideal. As described on page 28-29 in this study we employed AAV as merely a delivery tool for a proof-of-mechanism that underestimates the performance of an optimized viral vector system. Further, a non-viral mRNA mediated approach, such as lipid nanoparticle-mediated delivery, would be better suited for clinical translation as AAV promotes long-term expression, which is unnecessary for CRISPR/Cas9-mediated editing in the CNS.

1-2. *It would be additionally important to look at this fraction of double-positive neurons and quantify what fraction of such neurons lose pathologic hallmarks of repeat overexpression (DPRs, RNA foci).*

Since immunohistochemical detection of DPRs in BAC transgenic mice is inefficient and less quantitative than immunoassay detection strategies, we opted to employ the former and quantified total poly-dipeptide protein burden using well-characterized, highly-sensitive MSD immunoassays. This method is used extensively in the Brown (GP), Petrucelli and Gendron (GP), and Gao (GR) labs, and similar immunoassays are being used in clinical trials of G₄C₂ repeat-targeting therapies. We also explored measuring poly-dipeptide proteins using immunohistochemical approaches. Despite extensive optimization using flash-frozen tissues and multiple protocols, we have not succeeded in unequivocally identifying DPRs, either in untreated or treated transgenic C9 mice. While the reviewer is correct that quantifying RNA foci and DPRs specifically in double-positive neurons is conceptually elegant, we have found it to be technically impractical. Nevertheless, we were able to separately examine these two hallmark traits of c9ALS. As reported in the manuscript, we show that >50% of cells in treated striatum were transduced with both Cas9 and gRNA, and that this resulted in a >50% reduction of GP and GR, as well as >50% increase of nuclei without foci.

1-3. *The authors use PCR-based assays to assess editing efficiencies in a number of experiments (e.g. Fig 2b, fig 3b). These assays are non-linear and only semi-quantitative at best. In addition, it is now recognized that on-target genome rearrangements occur at unexpectedly high frequencies in iPSCs during Cas9-mediated editing, and if not directly assayed are often missed (PMID 30837594). Improved quantification and confirmation of editing through southern blotting in at least one of their cellular models (ideally patient-derived human iPSC neurons or organoids) would be superior.*

We agree that end-point PCR-based assays, while convenient, are limited in their ability to evaluate gene editing efficiencies and outcomes. Unfortunately, this is also true for a southern blotting. There are currently two published sets of probes for Southern blotting C9ORF72. One, often used in the Brown lab, probes five GGGGCC repeats. Because we are excising the repeats, this method would only be useful insofar as it documented a failure to detect repeats after excision therapy. The other probe targets the sequence upstream of the repeat expansion (241bp probe, DeJesus-Hernandez et al, 2011). Using this probe, we were able to see the WT and expanded band very faintly in human samples but not in the mouse samples. We sent some of our samples to colleagues in the Ranum lab, who were also unable to produce useful Southern Blot data.

To address the limitations of these techniques, we now employ two powerful approaches, UniDirectional Targeted Sequencing (UDiTaS, PMID: 29562890) and PacBio No-Amp sequencing to assess the efficiencies and outcomes of on-target genome editing.

1-4. *As a complementary technique, NGS sequencing of amplicons generated through primers flanking the edited site would enable them to identify allele-specific SNPs. Identification of allele-specific SNPs would allow the authors to distinguish between desired editing events and large “on-target” rearrangements in clones that could mimic the desired edit via gel electrophoresis but actually be due to deletion of primer binding sites on one of the alleles. An example of such an odd rearrangement may be clone 17, Fig 4b, but other clones that appear to be successfully edited based on gel electrophoresis could theoretically also harbor on-target rearrangements.*

As suggested by the reviewer, the PCR strategy was useful for screening iPSC clones and estimating editing efficiency but could yield false-positives. We therefore now include PacBio and UDiTaS analyses to validate the edited iPSC clone.

Incomplete evaluation of iPSC/iPSC neuron editing events and non-clonality assayed iPSC “clones”

1-5. *Testing their AAV system in patient-derived iPSCs with a single copy of bona-fide C9ORF72 repeats at the endogenous locus are key experiments for this manuscript. However, in Figure 4, the authors only demonstrate excision of the repeat in a single subclone (11-11), and in fact demonstrate that a “clone” in Fig4b was clearly polyclonal. In my opinion, this is likely because their protocol for isolating iPSC “clones” is highly biased toward selecting polyclonal clusters of cells. In their methods, they appear to pick clones post-dissociation after 72h. The doubling time of most iPSC lines is ~12-24h, so it is virtually impossible to pick a true clone of isogenic iPSCs 3d post-dissociation.*

We agree that polyclonal iPSC colonies were present during early stages of purification. However, after three rounds of titration, we were able to obtain purified sub-clones, as assessed by PCR with Sanger sequencing of amplicons, as well as repeat-primed PCR to confirm loss of the G4C2 expansion. Additionally, we also used immunostaining for PR to confirm loss of DPR expression and also found that editing rescues DNA damage phenotypes PMID: 32093728. As noted above in the new submission we have now used PacBio and UDiTaS to further assess proper editing in iPSC clones. We have now clarified the methods section to indicate that sub-clones were isolated after three rounds of colony selection after serial dilution.

1-6. *I would recommend that they repeat experiments performed in Fig 4b-d, modifying their protocol to improve frequency of isolating true single clones. The latter could be achieved through improved dissociation, FACS-based sorting of single cells into wells, or serial dilutions of dilute cell populations – coupled with waiting for a longer period of time prior to picking single clones – to improve their chances of identifying isogenic clonal populations. This would also allow them to assay additional clones (which I think is necessary) to demonstrate changes in poly GP levels and repeat prime PCR assays.*

Without a fluorescent marker to indicate proper genomic editing, the FACS-based approach would only increase screening efficiency by identifying GFP expressing cells that were transduced with the gRNA AAV vector and not cells transduced by the Cas9 AAV vector. FACS would also reduce efficiency due to cell injury and death associated with dissociating iPSC colonies and subjecting them to flow cytometry.

Our strategy optimized iPSC survival but entailed the drawback that serial dilution and selection of multiple sub-clones is a laborious process and quickly becomes unmanageable. For example, 3 subclones from each of the 24 original colonies yields 72 lines; an additional round of sub-cloning (3/line) yields 216 iPSC lines to maintain and screen. Because we assessed two gRNA sets in this study, there are therefore 432 lines. Here, for the first round of colony selection after transduction, we found evidence of editing in 13/24 colonies for gRNA2,3 (with clone 17 having the wrong band size, figure 4b). The gRNAs 2,4 only produced 6/24 colonies with evidence of editing. Our detecting multiple bands for several colonies indicates there are both edited and unedited cells in the populations assayed, as the

reviewer rightfully points out. We have therefore edited the text to distinguish “colonies” from the first two rounds of selection from purified “clones” isolated after 3 rounds of sub-cloning for this reason. Additionally, we analyzed our edited iPSC clone using PacBio and UDiTaS and confirmed the purity of the isogenic clone.

Minor points:

1-7. *The authors should note in the discussion that AAVs will not target certain cell types that may nevertheless be relevant to C9 pathogenesis, for example microglia.*

This is an important consideration. In the discussion on page 28-29 we describe that in this study we employed AAV as merely a delivery tool for a proof-of-mechanism. A non-viral mRNA mediated approach, such as lipid nanoparticle-mediated delivery, would be better suited for clinical translation as AAV promotes long-term expression, which is unnecessary for CRISPR/Cas9-mediated editing in the CNS. Lipid nanoparticles also have proven to be able to target microglia (PMC8815040).

1-8. *The editing frequency in organoids is modest at best; this should be pointed out. Appears likely due to relatively poor AAV transduction efficiencies based on GFP, likely compounded by issues that I raised regarding dual-transductions in single cell (if frequencies for the single GFP AAV are low, the chances of dual transduction are likely even lower).*

We agree that transduction efficiency is low. This likely reflects both the dual-transduction issue noted above and also AAV toxicity, which is pronounced in neural progenitors. However, we would like to point out that there are few studies that have optimized AAV transduction of brain organoids (MOI, serotype, etc). While our ongoing studies will establish optimal conditions, our data in this study are an underestimation of the potential editing efficiencies that could be achieved in the future. In our view, the main point is that we detected editing in the organoids, which provides additional proof of concept for the therapeutic approach. While AAV-mediated genome editing was demonstrated in iPSCs, iMNs, and brain organoids, optimizing editing efficiencies in each system was not pursued.

1-9. *The authors should consider mentioning that new versions of Cas9 and Cas9-like nucleases are much smaller than traditional spCas9, and that use of these nucleases could enable single AAV vector delivery of nucleases and sgRNAs. Eg. PMID: 32675376*

We have now included text indicating there are opportunities to improve upon our viral vector approach, including the use of alternative Cas9 variants with gRNAs in a single-vector.

1-10. *Fig 4h y-axis text is cut-off.*

The Figure 4h y-axis is now shown in full.

Reviewer #2

Meijboom et al developed AAVs to excise the pathogenic repeat expansion in C9orf72 models using CRISPR. They validated their approach in primary neurons, iPSC models and most importantly, also directly in two different BAC transgenic mouse models. This could be a very interesting therapeutic option in the future. The manuscript is well written and the data seems solid. For publication in Nat Comm some issues need to be addressed. Quantifying the efficacy of gene editing especially in mice seems most relevant.

We appreciate the encouraging statement by the reviewer and have now made additional efforts to assess the efficacy of gene editing. Notably, there is a dearth of validated neurodegenerative phenotypes in C9ORF72 BAC models, we are thus relegated to evaluating therapeutic efficacy by the reversal of molecular hallmarks (RNA foci, DPRs etc.)

Major points:

2-1. *Fig 2a: Homozygous x WT does not fit to the experiments with Cas9 transgenic mice.*

Cas9 transgenic embryos were generated in the same way as Fig. 2a: homozygous BAC111 mice (WT for Cas9) were crossed with homozygous B6J.129(Cg)-*Gt(Rosa)26Sor^{tm1.1(CAG-cas9*,-EGFP)Fezh/J}* (WT for the C9ORF72 transgene) resulting in heterozygosity of both BAC111 C9ORF72 and Cas9 embryos. We added this text to the legend of figure 2a, to clarify.

2-2. *Line 151: The authors used an MOI of 50000 in Cas9 expressing neurons. This high MOI will certainly not be achievable in patients. To get more therapy-relevant information the MOI should be titrated and editing efficacy quantified.*

We agree that this MOI is quite high, but we generated the *in vitro* data as proof of concept. For *in vivo* treatment we injected 1.2×10^{10} vector genomes (VG) of AAV9-Cas9 and 7×10^9 VG of the gRNA vectors or Rosa control per striatal side, which is 3.8×10^{10} VG in total for both striata per mouse and 1.65×10^{12} vector genomes per kilogram (we have added this to the methods on page 32). For patients, we would not inject the striatum, so it is hard to compare. But if we were to compare the dose that we gave our mice to the recommended dose of Zolgensma (an FDA approved AAV9 gene therapy for SMA patients), we see that their recommended dose is 1.1×10^{14} VG/kg (<https://www.fda.gov/media/126109/download>), which is much higher than the 1.65×10^{12} VG/kilogram we gave to our mice.

2-3. *Fig 2c: non-transgenic controls should be included to determine the background level and thus the extent of rescue. It would also be interesting to the community to compare the poly-GP level ins BAC111 and C9-500 neurons.*

Non-transgenic controls are shown in Figure 2c. While we agree that comparing poly-GP among mouse models would be interesting. This comparison would unfortunately be

complicated by the fact that the GP assessments were made in different labs using different antibodies. Nevertheless, we are encouraged by the consistency among findings that emerged from different models and assays.

2-4. *Fig 2d: Since foci are barely visible in the PBS BAC111 mice, non-transgenic controls should be shown. Are the foci only enlarged with gRNA2,3 or also gRNA2,4? This finding is very hard to explain. Is it associated with more intronic RNA on qPCR level? Were enlarged foci also observed in the treated BAC111 mice?*

We have changed the representative pictures (Figure 2d) into pictures that more clearly show the foci (previous picture showed many foci in the PBS treated BAC111 samples, but they were hard to see because the picture was so small). We have also added arrows and circles for the location of the RNA foci, in case they are still hard to see.

We also saw enlarged foci in treated BAC111 mice, but not in treated C9-500 mice and primary neurons.

While doing FISH for C9-500 primary neurons and striata, we also stained non-transgenic mouse primary neurons and striata and while we clearly saw foci in C9-500 mice, we did not see any foci in non-transgenic mice (we added pictures to supplementary data Figure 3).

2-5. *Fig 2g: The original western blots should be shown, because C9orf72 antibodies are notoriously bad.*

We have added pictures of the Western blots to supplementary Figure 2. We used the Proteintech polyclonal antibody 22637-1-AP for these experiments, which is extensively tested and compared to other C9 antibodies in PMID 28766957.

2-6. *Fig 2b/3b: Can the edited amplicon be quantified using real-time PCR to see the relative amount of editing under different conditions? Using unedited and edited iPSC lines from Fig 4 as a standard it might be possible to even quantify editing efficacy in absolute numbers.*

We designed two ddPCR probe sets. One probe binds to intron 1. If correct editing occurs (or insertions and inversions), this probe would not be able to bind to the gDNA. Using this method, we see >50% editing in gRNA2,3 and gRNA2,4 treated C9-500 and BAC111 striatal gDNA. The other probe has a forward primer just before the gRNA2 site and the reverse primer after the gRNA3/4 site. We used the edited iPSC line as a standard. However, if there are any insertions or extra deletions (which should not be an issue for the project as we are editing in the intron) the probe will not be able to bind anymore.

The other method of quantifying editing we used was UDiTaS (see Figure 4c).

ddPCR analysis of C9-500 striatal gDNA after AAV9-gRNA2,3 and AAV9-gRNA2,4 treatment using a ddPCR probe (**a.**) for the excised region intron 1, and (**b.**) one probe that has a forward primer just before the gRNA2 site and the reverse primer after the gRNA3/4 site. Expression levels were normalized to levels of a C9 probe at exon. Mean \pm SEM, n = 3-5, one-way ANOVA, Dunnett.

2-7. Fig 3c/d: Background in non-transgenic mice should be shown for GP and GR assays. The lack of correlation of GP and GR could be explained by poor performance of the GR assay. Have the GP and GR measurements been done on different animals to attribute it to injection issues? GP assays should be included for BAC111 mice. Absolute levels of GP/GR should be compared between the two mouse lines. Immunostaining should be shown to support immunoassay data.

Different animals were used for GP and GR measurements, as we need all protein of a whole striatum per MSD ELISA assay. We injected additional mice to repeat the GR assay in gRNA2,3-treated C9-500 mice, and found a clear correlation between GP and GR assay results (see Figure 3d).

Non-transgenic controls included in Figure 3c,d.

We opted to quantify poly-dipeptide proteins burden using well-characterized, sensitive MSD immunoassays. Not only is this method extensively used in the Brown lab (GP), the Petrucelli and Gendron labs (GP), and the Gao lab (GR), similar immunoassays are being used in clinical trials of G₄C₂ repeat-targeting therapeutics. We also chose to measure poly-dipeptide proteins by immunoassay rather than by immunohistochemical approaches because flash-frozen tissues (rather than paraffin-embedded tissues) are required for FISH and the detection of RNA foci. Despite extensive optimization using flash-frozen tissues to detect poly-dipeptide proteins by immunofluorescence or immunohistochemical staining (we tried 40+ different protocols), we were not successful. Nevertheless, we were able to separately examine these two hallmark traits of c9ALS. As mentioned before, we show that >50% of cells in treated striatum were transduced with both Cas9 and gRNA, and this resulted in a >50% reduction of GP and GR and a >50% increase of nuclei without foci.

2-8. Fig 3: Immunostaining for Cas9 should be shown to estimate transduction and spread in the CNS. The authors should attempt to quantify the percent of edited cells in vivo to support therapeutic utility. This might be done using qPCR or maybe DNA FISH of the edited (or unedited) locus.

We used RNAscope to stain for Cas9 and GFP (expressed by the gRNA vectors) and we found that most cells in the striatum were transduced with both AAVs, 58% for gRNA2,3 and 69% for gRNA2,4 treated C9-500 mice (Figure 4a,b). This is in line with the decrease we found in toxic gain of function, polydipeptides (Figure 3c,d) and RNA foci (Figure 3e). In order to quantify editing efficiency of our strategy, we used UDiTaS (Figure 4c).

2-9. *In all qPCR experiment the authors infer changes in transcript V2 arguing it cannot be quantified because primers would interfere with the gRNA target sites. However, V2 can be easily quantified directly using primers targeting the 3' end of the mRNA (e.g. PMID: 24559645 or commercial probes). This should be done.*

Since the 3' end of V2 and V3 are exactly the same, we cannot distinguish between the both with a probe targeting the 3' end of V2. The paper the reviewer is referring to only makes a distinction between short (V1) and long (V2 and V3 together) transcripts.

Minor points:

2-10. *Line 252: poly-GP is the most abundant poly-dipeptide from the antisense strand, but not in general.*

2-11. *Line 274/275: at least V3 did change in the primary neurons (see Fig 2f)*

2-12. *Line 325: figure 4g?*

These three points are now corrected.

Reviewer #3

In the manuscript by Meijboom KE et al., the authors used CRISPR-Cas9 system to excise expanded hexanucleotide repeats in C9ORF72 gene responsible for ALS/FTD. They demonstrated successful editing of this locus in various cellular models (human and mouse), brain organoids as well as in three mouse models. As a result the RNA foci and toxic polypeptides were reduced, and the transcript and protein levels were restored. The authors suggest that this strategy can be used as a therapeutic approach for ALS and FTD.

3-1. *What are the noteworthy results? The efficient excision of the mutant repeat tract with the use of CRISPR-Cas9 system has been already demonstrated for different STR-containing loci, including DMPK with a CTG expansion, HTT with a CAG expansion and even C9ORF72; therefore the approach presented in the manuscript is not original.*

For as far we are aware, there are no publications in which treatment with both gRNAs and Cas9 was done *in vivo*, or that utilize AAV vectors to deliver both the Cas9 and the gRNAs. The later noteworthy due to the increasing clinical utility of AAV-based drugs.

3-2. *The authors used the most basic CRISPR-Cas9 system tools, such as Streptococcus pyogenes Cas9 and a pair of unmodified gRNAs. Currently a genome editing field offers more advanced tools, more specific, safer, or smaller which would be better for therapeutic approaches. Although promising, the therapeutic application of the strategy presented in the manuscript would require more research including more detailed analysis of off-targets, improving delivery issues, etc.*

Conventional SpCRISPR-Cas9 can be nonspecific and further improvements of this strategy are necessary. In addition the author's findings that excision of expanded repeats rescues major disease mechanisms in vitro is not completely novel. For example, Selvaraj et al., (doi: 10.1038/s41467-017-02729-0) demonstrated that excision of the G4C2 repeat expansion by CRISPR-Cas9 system reverses RNA foci pathology in mutant motor neurons.

This paper used nucleofection, not AAV. Ours is a proof of concept for an AAV based therapy in vivo. The paper actually supports our approach. Additionally, in this paper only the gain of function disease mechanisms were rescued (RNA foci and polydiptides) but C9 transcript and protein levels were not measured/rescued. Lastly, this paper shows no *in vivo* data. It is encouraging for the development of future AAV-based drugs that we were able to rescue molecular phenotypes of disease despite using less sophisticated CRISPR tools.

Nonetheless, it is an interesting study which extends other reports and demonstrates that the CRISPR-Cas9 technology can be successfully used to decrease pathological hallmarks of ALS/FTD. The manuscript is well written, the data are generally strong and well presented. The number of models is large. However, some conclusions are not supported by the data and some aspects needs clarification:

Major points:

3-3. *Additional evidence is needed to support a conclusion that the presented strategy does not generate off-target effects. In the paper “Unexpected Mutations by CRISPR-Cas9 CTG Repeat Excision in Myotonic Dystrophy and Use of CRISPR Interference as an Alternative Approach” (DOI: 10.1016/j.omtm.2020.05.024) the authors used similar strategy to excise long CTG repeats in DMI. The authors used a conventional Cas9 nuclease and Cas9 nickase and demonstrated that the formation of the RNA foci was markedly reduced in patient-derived fibroblasts. In addition they observed a considerable amount of unintended off-target mutations and genomic rearrangements using high-throughput genome-wide translocation sequencing. Therefore, more detailed analysis of unintended events (especially genomic rearrangements) should be performed in this study using unbiased method to support the idea of using this approach for therapeutic purposes. The authors assumed that excision of ex2 present in V2 mRNA (the most highly expressed C9ORF72 variant) has no influence on the overall mRNA and protein levels. Perhaps it is so, but a more “elegant” approach would be the excision of repeats without disruption of the ORF. Was it impossible ? (splicing disruption, lack of a PAM sequence within 35bp flanking sequence,...)?*

The 35bp flanking sequence consist for 94% of guanines and cytosines, only 2 single bp are thymines. When designing gRNAs, you ideally want to stay in the 40-60% range of GC. Furthermore, this 35 bp sequence is highly represented throughout the genome, so any gRNA designed for this area has an extreme high prediction of off-targets. We agree that it would be a more elegant approach- if possible- but luckily, we have not seen a negative effect on transcript or protein levels, which can be explained by the fact that exon 2 is non-coding. We also added a detailed analysis of unintended editing events, using UDiTaS in HEK293T cells (Figure 1e), patient iPSCs (Figure 5f) and *in vivo* (Figure 4c).

3-4. *The authors used biased PCR-based methods to analyze the quality and quantity of edits. This excludes the possibility of detection of larger deletions, expansions.*

We used No-Amp PacBio and UDiTaS to analyze the quality and quantity of edits, which made it possible to detect larger deletions and expansions (Figures 1e, 4c, 5e, 5f).

3-5. *Efficient excision was demonstrated using many different models. Did the authors expect different DNA repair results in these models? (different quality or quantity of edits?).*

We did not expect a different DNA repair results in different models, but we wanted to prove beyond doubt that our therapeutic strategy is not only able to excise the repeat expansion, but also thereby rescuing C9ORF72 disease mechanisms. This is for as far the authors know, currently the only study that shows a rescue of all three C9 disease mechanisms (RNA foci, toxic polydipeptides and haploinsufficiency) *in vivo* and *in vitro*. Although we expected similar DNA repair results in different models, we were not sure if each model would respond the same to this DNA repair.

3-6. *Fig.2d: how to explain that the number of foci is higher in BAC111 model treated with gRNA 2,3? The conclusion presented by the authors (lines 199-201) is not convincing, since the control cells (PBS) do not have foci (at least in the picture shown).*

Apologies for the small picture. At better quality, it shows many RNA foci in the PBS treated BAC111 nuclei, but they were hard to see because the picture was so small). We have changed the representative pictures (Figure 2d) into pictures that more clearly show the foci. We have also added arrows and circles for the location of the RNA foci in case they are still hard to see.

3-7. *Fig.3d: as above, I am not convinced by the explanation of poly-GR levels in C9-500 mice treated with gRNA 2,3 (lines 263-266). The authors did not exclude the possibility of genomic rearrangements, or other unintended events induced by excision of a large DNA sequence. In addition, since RAN proteins production can be higher in stress conditions and AAV9-CRISPR-Cas9 system injection induces stress, PBS control is probably not the best choice for this experiment.*

We injected additional mice to repeat the GR assay in gRNA2,3-treated C9-500 mice, and found a clear correlation between GP and GR assay results (see Figure 3d).

In order to examine the possibility of genomic rearrangements, or other unintended events induced by excision of a large DNA sequence, we added UDiTaS analyses (Figures 1e, 4c, 5f).

Regarding the reviewer's reasoning that the PBS control would not be the best choice for this experiment, because PBS treatment could induce stress and thereby cause a higher RAN protein production. We did not think this would be an issue as we harvested striata 8 weeks after injection and regarding treatment of primary neurons, the addition of 5ul of PBS into 2ml of media should not stress the primary neurons. Since AAV-mediated stress would

increase DPR production, the finding that treatment reduced DPRs speaks to the efficacy of the vectors, despite the potential role of stress-induced DPR production.

Minor points:

3-8. *Dividing the text into sub-chapters would be helpful especially when so many different models are used.*

We followed Nature Communication guidelines.

3-9. *Fig. 1b (lines 117-119), it is unclear if control was Untr (?) or plasmid, please explain*

Apologies, we used both an untreated control and a control in which we treated cells with a GFP expressing plasmid. I changed Figure 1b. and the text.

3-10. *Fig. 1 description: NoE-F1 primer is not present on the Figure and its function is unclear*

Corrected

3-11. *Line 92: please expand the abbreviation iMN,*

Corrected

3-12. *Line 121: “efficient” instead of “efficiently”*

Corrected

3-13. *Line 126, 136: “sanger sequencing”, “sanger-sequencing”*

Corrected

3-14. *Lines 129-130: please correct the sentence*

Corrected

3-15. *Line 325: Fig.4g instead of 4e*

Corrected

3-16. *Line 544: What does it mean that InDels up to 20bp were permitted? Please explain.*

With this method we only detected InDels that were not larger than 20bp. With the UDiTaS data we also investigate larger InDels.

3-17. *Line 551: please explain why AMACR locus was analyzed using different parameters; please add more details concerning off-target analysis and deep sequencing (the length of amplicons, depth, RPM,...)*

Typically, amplicons are designed to be ≤ 290 bp with ≥ 10 bp of R1/R2 overlap sequenced 150 cycles from each end. AMACR had the only amplicon for which such overlap was not achievable due to priming difficulties in this region. We therefore selected R2 which fully encompasses our sgRNA binding site and adjacent nucleotides for analysis in single-end mode. Running this amplicon in paired-end mode would likely have unknown consequences on the analysis output due to lack of overlap. We cannot find any significant drawbacks of running single-end mode noted in the software description or literature.

We have added the off-target analysis to the supplementary data for more details.

REVIEWERS' COMMENTS

Reviewer #1 (Remarks to the Author):

This is a review of the resubmitted manuscript by Meijboom entitled "CRISPR/Cas9-Mediated Excision of ALS/FTD-Causing Hexanucleotide Repeat Expansion in C9ORF72 rescues major disease mechanisms in vivo and in vitro". The authors have now performed extensive additional experiment to characterize an AAV CRISPR/Cas9 gene editing strategy for excision of mutant C9 repeats. My earlier concerns focused mainly on technical aspects of the work and related caveats to interpretation of resulting data, which I feel are largely now addressed in the revision.

In addition to these general comments, I would like to provide the additional feedback below and relatively minor questions for further consideration:

- The authors now show that co-transduction of Cas9 and sgRNAs by two AAVs are ~60-70% in mouse striatum, roughly in line with expectations on co-transduction efficiencies. The resulting rescue effects were roughly correlated with these co-transduction results, further supporting a conclusion that successful deletion of the C9 repeat normalized the phenotype in these dual-transduced cells. The authors were unable to assess rescue effects on an individual neuron basis due to technical concerns, despite substantial efforts on their part; this limitation should be noted in the discussion section.
- The authors further characterize possible outcomes of the editing process using long read sequencing techniques, which was an essential new experiment that provides further rigor and nuanced findings. Of note, they see substantial integration of AAV genomes at the edited site in certain contexts, which is certainly of importance in the context of consideration of AAV-mediated delivery of CRISPR/Cas9 in settings of therapeutic application, strongly suggesting that safer non-AAV strategies will be needed. The authors appropriately note this and other AAV-related caveats in their new version of the manuscript.
- I am surprised that the authors do not appear to see any on-target deletions of regions of the genome adjacent to their targeted edited site, in contrast to published reports from multiple teams (PMID 32460021 and Skarnes at [el biorxiv https://doi.org/10.1101/2021.12.16.472942](https://doi.org/10.1101/2021.12.16.472942)). It would be helpful if the authors could comment on this unexpected observation in their discussion section, and whether their primary long-read sequencing strategy (UDiTaS) has completely addressed this possibility. I note that they use a C9orf72 specific primer as part of this strategy; can they exclude possible outcomes in which the primer binding site was also deleted along with longer on-target deletion events? If their long-read sequencing captured allele-specific SNPs, they could likely also exclude such possibilities, but it is unclear to me whether they investigated allele specific SNPs as part of the UDiTaS sequencing strategies. I continue to raise this point because if on-target deletions do occur with dual sgRNA approaches to excise hexanucleotide repeats, those events could be quite destructive in the context of possible therapeutic uses, regardless of delivery technique of the CRISPR/Cas9 machinery.

Reviewer #2 (Remarks to the Author):

The manuscript improved significantly and is now ready for publication.

Reviewer #3 (Remarks to the Author):

I would like to thank the authors for responding to my comments and performing additional experiments that improved the quality of the research.

I only have one comment regarding the point 3-7 (Fig.3d)

I didn't mean that PBS induced stress in cells. Injection of the AAV9-CRISPR-Cas9 viral vector may induce stress, which may result in an increase in aggregates, so perhaps better control would be the scramble construct (AAV9-CRISPR-Cas9-Scr). However, I agree with the authors' explanation that since treatment reduces aggregates, it is in favor and confirms the validity of this strategy.

Reviewer #1 (Remarks to the Author):

This is a review of the resubmitted manuscript by Meijboom entitled “CRISPR/Cas9-Mediated Excision of ALS/FTD-Causing Hexanucleotide Repeat Expansion in C9ORF72 rescues major disease mechanisms in vivo and in vitro”. The authors have now performed extensive additional experiment to characterize an AAV CRISPR/Cas9 gene editing strategy for excision of mutant C9 repeats. My earlier concerns focused mainly on technical aspects of the work and related caveats to interpretation of resulting data, which I feel are largely now addressed in the revision.

Thank you for reviewing our manuscript, we appreciated your suggestions for improving the quality of our manuscript!

In addition to these general comments, I would like to provide the additional feedback below and relatively minor questions for further consideration:

- The authors now show that co-transduction of Cas9 and sgRNAs by two AAVs are ~60-70% in mouse striatum, roughly in line with expectations on co-transduction efficiencies. The resulting rescue effects were roughly correlated with these co-transduction results, further supporting a conclusion that successful deletion of the C9 repeat normalized the phenotype in these dual-transduced cells. The authors were unable to assess rescue effects on an individual neuron basis due to technical concerns, despite substantial efforts on their part; this limitation should be noted in the discussion section.

We have now mentioned this on page 27.

- The authors further characterize possible outcomes of the editing process using long read sequencing techniques, which was an essential new experiment that provides further rigor and nuanced findings. Of note, they see substantial integration of AAV genomes at the edited site in certain contexts, which is certainly of importance in the context of consideration of AAV-mediated delivery of CRISPR/Cas9 in settings of therapeutic application, strongly suggesting that safer non-AAV strategies will be needed. The authors appropriately note this and other AAV-related caveats in their new version of the manuscript.

- I am surprised that the authors do not appear to see any on-target deletions of regions of the genome adjacent to their targeted edited site, in contrast to published reports from multiple teams (PMID 32460021 and Skarnes at *el biorxiv* <https://doi.org/10.1101/2021.12.16.472942>). It would be helpful if the authors could comment on this unexpected observation in their discussion section, and whether their primary long-read sequencing strategy (UDiTaS) has completely addressed this possibility. I note that they use a C9orf72 specific primer as part of this strategy; can they exclude possible outcomes in which the primer binding site was also deleted along with longer on-target deletion events? If their long-read sequencing captured allele-specific SNPs, they could likely also exclude such possibilities, but it is unclear to me whether they investigated allele specific SNPs as part of the UDiTaS sequencing strategies. I continue to raise this point because if on-target deletions do occur with dual sgRNA approaches to excise hexanucleotide repeats, those events could be quite destructive in the context of possible therapeutic uses, regardless of delivery technique of the CRISPR/Cas9 machinery.

We did indeed not see any on-target deletions of regions of the genome adjacent to the targeted edited site in the isogenic iPSC cell line. We confirmed this UDiTaS result (Figure 5f) with PacBio no-AMP sequencing (for iPSCs we had more reads, so we were able to analyze it, see

Figure 5e). This clean editing likely reflects the way the treated isogenic iPSC line was created, by picking clones and subclones after treatment. All cells should have the same edit.

However, using UDiTaS to analyze editing in gRNA2,3- and gRNA2,4-treated C9 mouse striata, we did see on-target (small) deletions (see Figure 4d, green bar). Because of methodological limitations of UDiTaS and our use of a single locus-specific primer, we are not able to see large insertions or deletions (<140 bp) or deletions of the primer binding site. We have added a statement about this limitation on page 14 and we referenced PMID 32460021. When looking at PacBio data we found only one read in all treated samples with a deletion that was large enough to take out the primer binding site (which was 0.3% of total reads combined of gRNA2,3 and gRNA2,4 treated samples, both PacBio No-AMP rounds).

We performed two rounds of PacBio No-AMP sequencing on DNA extracted from striatal tissues, but found the number of reads was insufficient to detect rare editing events. We also performed UDiTaS to gain more insight into editing outcomes. Despite these efforts, the reviewer is correct that some editing events, including on-target deletions may not have been detected in our analyses. We have acknowledged the limitations of current technologies to uncover all potential editing outcomes. With that said, editing within introns (C9ORF72 intron 1), insertions and deletions are less relevant, because they are not likely to change amino acid composition of the C9ORF72 protein and are unlikely to change transcription levels, even if editing occurred in the WT allele, so those editing events should not be destructive. For this project it was more relevant - next to quantifying HRE excision/deletion - to quantify inversions, as inversions of the HRE would still likely cause RNA foci and RAN translation into toxic polypeptides. However, potential inversions of the HRE in the WT allele should not cause RNA foci and RAN translation, as the HRE is too small.

Since our editing strategy does not discriminate between expanded and WT alleles (as mentioned on page 5), both PacBio and UDiTaS are unable to distinguish between allele-specific SNPs, after editing. The iPSC and iMN line we used for editing expresses both a WT and an expanded allele. We did not see any negative effects of editing on mRNA and protein levels (quite the opposite, we saw a strong significant increase of both, Figure 5 j,k).

Reviewer #2 (Remarks to the Author):

The manuscript improved significantly and is now ready for publication.

Thank you for reviewing our manuscript, we appreciated your suggestions for improving the quality of our manuscript!

Reviewer #3 (Remarks to the Author):

I would like to thank the authors for responding to my comments and performing additional experiments that improved the quality of the research.

I only have one comment regarding the point 3-7 (Fig.3d)

I didn't mean that PBS induced stress in cells. Injection of the AAV9-CRISPR-Cas9 viral vector may

induce stress, which may result in an increase in aggregates, so perhaps better control would be the scramble construct (AAV9-CRISPR-Cas9-Scr). However, I agree with the authors' explanation that since treatment reduces aggregates, it is in favor and confirms the validity of this strategy.

Apologies for the misunderstanding and thank you for the explanation! We will keep these suggestions in mind for future research. Thank you for reviewing our manuscript, we appreciated your suggestions for improving the quality of our manuscript!